# FineCLIP: Self-distilled Region-based CLIP for Better Fine-grained Understanding

**Dong Jing**[*1]**, Xiaolong He**[*1]**, Yutian Luo**[1]**, Nanyi Fei**[2]**,**
**Guoxing Yang**[1]**, Wei Wei**[3]**, Huiwen Zhao**[3]**, Zhiwu Lu** [†1]
[1]Gaoling School of Artificial Intelligence, Renmin University of China
[2]MetaBrain AGI Lab, Shanghai, China
[3]R&D Management Department, Honor Device Co., Ltd
`{jingdong98, xiaolonghe, luzhiwu}@ruc.edu.cn`

## Abstract

Contrastive Language-Image Pre-training (CLIP) achieves impressive performance on tasks like image classification and image-text retrieval by learning on large-scale image-text datasets. However, CLIP struggles with dense prediction tasks due to the poor grasp of the fine-grained details. Although existing works pay attention to this issue, they achieve limited improvements and usually sacrifice the important visual-semantic consistency. To overcome these limitations, we propose FineCLIP, which keeps the global contrastive learning to preserve the visual-semantic consistency and further enhances the fine-grained understanding through two innovations: 1) A real-time self-distillation scheme that facilitates the transfer of representation capability from global to local features. 2) A semantically-rich regional contrastive learning paradigm with generated region-text pairs, boosting the local representation capabilities with abundant fine-grained knowledge. Both cooperate to fully leverage diverse semantics and multi-grained complementary information. To validate the superiority of our FineCLIP and the rationality of each design, we conduct extensive experiments on challenging dense prediction and image-level tasks. All the observations demonstrate the effectiveness of FineCLIP.

## 1 Introduction

Contrastive Language-Image Pre-training (CLIP) [18, 40] emerges as the foundational work in vision-language representation learning. By training on large-scale, noisy image-text pairs, CLIP aligns global image and text embeddings within a unified latent space, demonstrating remarkable successes across image-level tasks [1, 25, 61], e.g. image classification and cross-modal retrieval.

However, CLIP has shown notable limitations in understanding fine-grained details, such as identifying object attributes and their relationships [37, 41, 62, 70]. Especially, when applied to downstream tasks, CLIP struggles to extract valuable region representations from visual dense features, limiting its effectiveness in complex recognition scenarios [33, 72]. Recent works [60, 70] attribute this issue to the task domain shift: CLIP matches an image as a whole to text description but fails to capture fine-grained alignment between image regions and corresponding textual attributes.

To address this problem, researchers have attempted to enhance fine-grained alignment using two primary strategies. The first strategy [27, 59, 66, 70] directly leverages CLIP to match image regions with template labels using large quantities of grounding annotations in a classification setting. However, the pre-defined template labels lack sufficient semantic diversity, restricting its

---

[*]Equal Contribution
[†]Corresponding Author

generalization to open-world scenarios. The second strategy [55] proposes a uni-modal distillation scheme by aligning the region dense features of the trainable student model with the image-level representation of the corresponding image crops generated by the frozen teacher model. Although efficient, the frozen teacher model restricts the performance ceiling of the student model. Notably, both of the strategies disrupt the important semantic consistency of visual representations.

In this work, we unify cross-modal regional alignment and uni-modal global-to-region guidance into a coherent framework. We present **FineCLIP**, an end-to-end universal vision-language framework that gains better fine-grained understanding by reasonably incorporating a multi-grained contrastive learning paradigm with a real-time self-distillation scheme. FineCLIP involves the following appealing designs: **1)** In order to enrich the model with abundant and diverse fine-grained semantics, instead of using limited template labels in the classification setting, we build the **regional contrastive learning paradigm** using regions and corresponding text descriptions generated by advanced Large Vision-Language Model (LVLM). **2)** To facilitate interactions between global embeddings of image region crops and corresponding region dense features for mutual guidance, we introduce a **real-time self-distillation scheme** that relies on and complements global contrastive learning. Instead of using the frozen teacher model for guidance, we use the trainable FineCLIP to independently teach itself, leading to improved performance as global representations are progressively refined during training. **3)** FineCLIP universally aligns visual embedding, visual dense features, and textual embedding into a unified latent space. With these design considerations, FineCLIP fully leverages available diverse semantics from fine-grained region descriptions and real-time optimal global representations, and boosts the interactions of multi-grained complementary information. When applied to downstream tasks, FineCLIP is capable of effectively handling both dense prediction using region dense features and image-level tasks using semantic-aligned global embeddings.

Through extensive experimental evaluations, we show that FineCLIP surpasses previous arts on most dense prediction tasks and image-level tasks under fair comparison settings, demonstrating its effectiveness in both fine-grained understanding and semantic-aligned global representation. Furthermore, FineCLIP presents promising scaling ability, consistently showing faster performance improvements than other competing methods as we scale up the trainset.

Our contribution is threefold: **1)** We present FineCLIP, which clearly incorporates the multi-grained contrastive learning paradigm and the real-time self-distillation scheme to achieve better fine-grained understanding. **2)** We develop an automated region-text data generation pipeline utilizing advanced LVLMs, demonstrating its effectiveness in providing valuable fine-grained semantics. **3)** Extensive experiments on dense prediction and image-level benchmarks show that our FineCLIP consistently outperforms previous arts and exhibits promising scalability.

## 2 Methodology

### 2.1 Preliminary

**CLIP Architecture.** CLIP is a dual-tower architecture composed of a vision encoder $\mathcal{V}$ (ViT [9]) and a language encoder $\mathcal{L}$ (BERT [8]). Given an image-text pair, CLIP outputs the visual $[CLS]$ token $v$, visual dense features $\mathcal{X}$ corresponding to image patches, textual $[CLS]$ token $t$ and textual dense features of text tokens. The $[CLS]$ tokens $v$ and $t$ serve as global image and text embedding, respectively. During the pre-training, the global contrastive loss is computed with $v$ and $t$ for instance-level alignment. In downstream applications, the global embeddings $v$ and $t$ are crucial in image-level tasks such as image classification and image-text retrieval, whereas visual dense features $\mathcal{X}$ are vital for dense prediction tasks like object detection and semantic segmentation.

**Problem Definition.** Our aim is to develop a comprehensive representation space where visual and semantic features are both globally and locally aligned, which contributes to create a robust vision-language model that can effectively address both image-level and dense prediction tasks.

To achieve the goal, the model must satisfy two key requirements at both global and regional levels: **1)** Given a text $T$ that describes the content of an image $I$, the image embedding $v$ should be matched to text embedding $t$. **2)** Given a regional text $T^r$ describing the content of a specific region $r$ within the image $I$, both the local visual feature $p^r$, pooled from visual dense features $\mathcal{X}$ according to $r$, and region visual embedding $v^r$ of region crop $I^r$ should align with region text embedding $t^r$ of $T^r$.

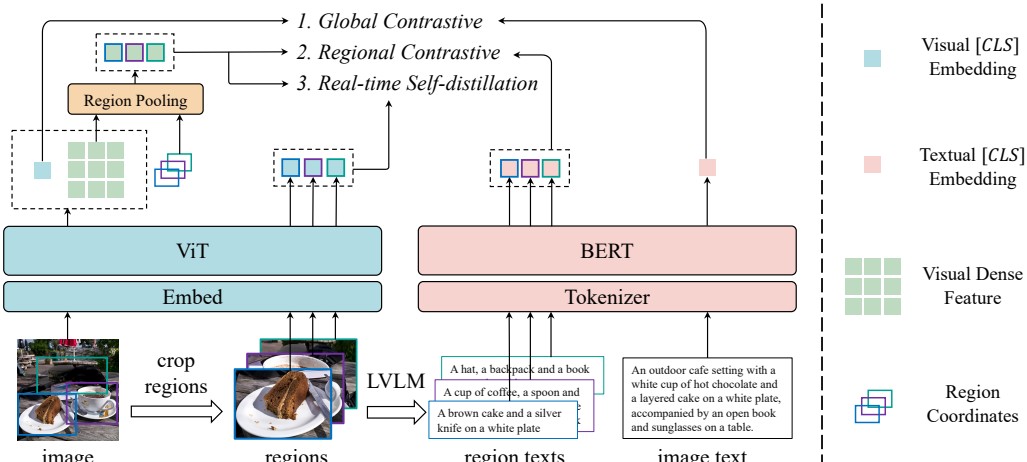

Figure 1: Overall architecture of FineCLIP. For simplicity, the diagram omits unused visual dense features of regions extracted by ViT and textual dense features generated by BERT. By integrating multi-grained contrastive learning as well as a real-time self-distillation scheme, FineCLIP aligns visual global embedding, regional dense features, and textual global embedding into a unified space, acquiring rich coarse and fine-grained knowledge from image-text and generated region-text pairs.

## 2.2 FineCLIP

**Overview.** As depicted in Figure 1, FineCLIP adopts the same architecture with CLIP, which consists of the vision encoder $\mathcal{V}$ and the language encoder $\mathcal{L}$, but employs more complex inputs and objectives.

The input batch of FineCLIP contains image-text pairs $\{I_i, T_i\}_{i \leq \mathcal{B}}$, region-text pairs $\{I_j^r, T_j^r\}_{j \leq \mathcal{M}}$ and a set of corresponding region coordinates $\{r_j\}_{j \leq \mathcal{M}}$, where $\mathcal{B}$ and $\mathcal{M}$ refers to the batch size and region count. Noted that the regions $\{I_j^r\}_{j \leq \mathcal{M}}$ are cropped from images $\{I_i\}_{i \leq \mathcal{B}}$ according to region coordinates $\{r_j\}_{j \leq \mathcal{M}}$, and region texts $\{T_j^r\}_{j \leq \mathcal{M}}$ are generated by LVLMs.

Recall the image and text processing of CLIP introduced in Section 2.1. For image-text pairs $\{I_i, T_i\}_{i \leq \mathcal{B}}$ and region-text pairs $\{I_j^r, T_j^r\}_{j \leq \mathcal{M}}$, FineCLIP outputs the corresponding global image-text embeddings $\{v_i, t_i\}_{i \leq \mathcal{B}}$ and $\{v_j^r, t_j^r\}_{j \leq \mathcal{M}}$, respectively. Moreover, FineCLIP extracts regional visual representations $\{p_j^r\}_{j \leq \mathcal{M}}$ by pooling visual dense features $\{\mathcal{X}_i\}_{i \leq \mathcal{B}}$ according to region coordinates $\{r_j\}_{j \leq \mathcal{M}}$ with RoIAlign [15].

**Global Contrastive Learning.** The global contrastive learning realizes the instance-level alignment, which enables FineCLIP to bolster multi-modal global embeddings and acquire rich coarse-grained knowledge. Initially, the cosine similarity between image embedding $v$ and text embedding $t$ is calculated as

$$S(v,t) = \frac{v \cdot t}{||v|| \, ||t||}. \tag{1}$$

The global contrastive loss forces FineCLIP to learn global image and text embeddings by maximizing the cosine similarity to the corresponding text and image embeddings, while minimizing the cosine similarity to other non-corresponding ones in the batch, i.e.

$$L_{GC} = -\frac{1}{2\mathcal{B}} \sum_{i=1}^{\mathcal{B}} (\log \frac{\exp(S(v_i, t_i)/\tau)}{\sum_{j=1}^{\mathcal{B}} \exp(S(v_i, t_j)/\tau)} + \log \frac{\exp(S(t_i, v_i)/\tau)}{\sum_{j=1}^{\mathcal{B}} \exp(S(t_i, v_j)/\tau)}), \tag{2}$$

where $\tau$ is the learnable temperature and initialized with $1e-2$.

**Real-time Self-distillation Scheme.** Since the global contrastive loss $L_{GC}$ supervises only global embeddings, it is poorly effective in improving local dense features.

The distillation scheme [55, 60], which transfers robust global representational capabilities to region features, emerges as a promising solution to this problem. The previous work of Wu et al. [55] implemented the distillation scheme by guiding a trainable student model's local feature extraction

with the global visual embeddings of corresponding image crops from a frozen teacher model. While efficient, this method depends on the pre-trained teacher model and does not support training from scratch. Additionally, because the student model aligns fully with the frozen teacher model, its performance is capped by the teacher's capabilities. This performance ceiling is quickly reached during pre-training, limiting further improvement and scalability, as illustrated in Figure 2.

We propose a real-time self-distillation scheme that relies on and complements global contrastive learning, eliminating the need for a frozen teacher model. Instead of relying on a frozen teacher model for providing high-quality global image embeddings, global contrastive learning consistently enhances the student's global representation capability during the training, allowing the student model to teach itself independently. Importantly, our implementation leverages real-time optimal global representations during the training for guidance, resulting in better scalability and fine-grained understanding ability of the student model.

Specifically, as shown in Figure 1, our real-time self-distillation loss directly maximizes the cosine similarity between region features $\{p_j^r\}_{j \leq \mathcal{M}}$ pooled from image dense features and the visual representations of region crops $\{v_j^r\}_{j \leq \mathcal{M}}$ using the formula

$$L_{SD} = \frac{1}{\mathcal{M}} \sum_{j=1}^{\mathcal{M}} (1 - S(p_j^r, v_j^r)). \tag{3}$$

**Semantically-rich Regional Contrastive learning.** Despite the remarkable enhancements through aforementioned designs, the model still lacks abundant fine-grained knowledge due to the utilization of coarse-grained training data.

Intuitively, to enrich the model with fine-grained knowledge, it is crucial for the model to focus more on the specific content of image regions and learn more precise and detailed semantics. Therefore, we are motivated to implement regional contrastive learning that operates on the level of regions and related descriptions. As shown in Figure 1, we leverage the advanced LVLM to generate high-quality region descriptions with diverse fine-grained semantics.

The regional contrastive loss compels FineCLIP to learn pooled visual region features and text embeddings by maximizing the cosine similarity between matching text and region pairs, while minimizing the similarity with non-matching pairs in the batch, which is defined as follows

$$L_{RC} = -\frac{1}{2\mathcal{M}} \sum_{i=1}^{\mathcal{M}} (\log \frac{\exp(S(p_i^r, t_i^r)/\tau)}{\sum_{j=1}^{\mathcal{M}} \exp(S(p_i^r, t_j^r)/\tau)} + \log \frac{\exp(S(t_i^r, p_i^r)/\tau)}{\sum_{j=1}^{\mathcal{M}} \exp(S(t_i^r, p_j^r)/\tau)}). \tag{4}$$

The reason for utilizing pooled visual region features instead of image embeddings of region crops is two-fold. **1)** Supervising pooled region features directly improves model's capability to extract valuable local dense features. **2)** Enhanced by attention mechanisms in ViT, pooled region features encompass a border perception of entire image content compared to image embeddings of regions, which facilitates a deeper semantic understanding.

**Learning Objective.** As depicted in Figure 1, the learning objective of FineCLIP incorporates the above three components. Global contrastive loss $L_{GC}$ works for enhancing representation capabilities and semantic consistency of global visual embeddings, while the self-distillation loss $L_{SD}$ is designed to transfer the real-time strong representation capability of global embeddings to local visual features. Additionally, regional contrastive loss $L_{RC}$ is applied to enrich the FineCLIP with fine-grained knowledge and further improve its local features. The learning objective is

$$L = L_{GC} + \lambda * L_{SD} + \gamma * L_{RC}, \tag{5}$$

where the $\lambda$ and $\gamma$ are hyper-parameters. As a result, FineCLIP fully leverages available diverse semantics and boosts the interactions within multi-grained complementary information.

## 3 Experiment

### 3.1 Ablation Study of FineCLIP

**Experiment Settings.** In our ablation experiments, we train FineCLIP using $8 \times$A800 GPUs on `train2017` split of COCO dataset [30], which includes approximately $118$K human-annotated

image-text pairs along with 970K region-label pairs. To provide abundant fine-grained knowledge, we replace labels provided by COCO with region descriptions generated by BLIP-2 [26]. FineCLIP is initialized by ViT-B/16 with default input image size of $224 \times 224$ and corresponding BERT from EVA-CLIP [49]. We train FineCLIP for 10 epochs using AdamW [32] optimizer with the batch size of 32 per GPU, the learning rate of $1e-5$, and the weight decay of 0.1. The coefficients $\lambda$ and $\gamma$ in learning objective are both set to 1. In all experiments, we freeze the language encoder $\mathcal{L}$ to reduce computational overheads and improve training stability.

Using the COCO `val2017` split, we test FineCLIP designs on the box classification task with pooled region features and image-level retrieval tasks using global embeddings. We report the Top1 and Top5 mean accuracy of all annotated boxes in the box classification task, and the R@1 accuracy of image-to-text and text-to-image retrieval tasks as evaluation indicators.

**Ablation of Objective Components.**
The training objective of FineCLIP includes three components: $L_{GC}, L_{SD}$ and $L_{RC}$. We first examine the combination of $L_{GC}$ and $L_{SD}$. In Table 1 (row) #1, using only $L_{SD}$ for supervision causes the model's accuracy on both box classification and retrieval tasks to drop to near zero. This training collapse is predictable, as the global representation ability of

Table 1: Ablation study on the objective components.

| # | $L_{GC}$ | $L_{SD}$ | $L_{RC}$ | Box Classification | | Retrieval | |
|---|---|---|---|---|---|---|---|
| | | | | Top1 | Top5 | I2T | T21 |
| 1 | | √ | | 0.0 | 0.0 | 0.0 | 0.1 |
| 2 | √ | | | 42.3 | 66.6 | 62.4 | 48.8 |
| 3 | √ | √ | | 43.7 | 72.9 | 60.0 | 47.1 |
| 4 | | | √ | 45.5 | 72.0 | 39.5 | 30.4 |
| 5 | √ | | √ | 47.8 | 74.1 | **62.5** | **48.9** |
| 6 | √ | √ | √ | **48.4** | **75.6** | 62.2 | 47.6 |

the model is entirely compromised without the supervision of $L_{GC}$ during the training. In Table 1 (row) #3, incorporating $L_{GC}$ to support global embeddings allows the training to proceed stably. Comparing Table 1 (rows) #2 and #3 reveals that while the self-distillation loss improves the box classification performance, it slightly reduces retrieval performance, leading to a trade-off between local feature extraction and global representations.

As shown in Table 1 (rows) #2 and #5, $L_{RC}$ significantly improves the model's region feature extraction (+5.5 of Top1 and +7.5 of Top5 mean accuracy on box classification) and slightly enhances retrieval performance. This improvement is attributed to the introduction of region-text pairs, which provide abundant fine-grained knowledge, and the positive effects of $L_{RC}$ on boosting local features. This result also indicates that, despite the possible noise in the region-text pairs generated by BLIP-2, they still offer valuable information for learning region representations. Ultimately, combining all three components enables FineCLIP to achieve optimal classification performance and competitive retrieval results, as demonstrated in Table 1 (row) #6.

Table 2: Performance comparisons of FineCLIP using different region proposal methods.

| # | Region Proposal Method | Box Classification | | Retrieval | | Num of Regions (per image) | Time Overhead |
|---|---|---|---|---|---|---|---|
| | | Top1 | Top5 | I2T | T2I | | |
| 1 | Manual [30] | 48.4 | 75.6 | **62.2** | **47.6** | 9 | - |
| 2 | FastSAM [69] | 47.1 | 73.7 | 60.7 | 46.5 | 23 | 15 min |
| 3 | RPN [43] | 48.8 | 76.0 | 61.5 | 46.9 | 16 | 10 min |
| 4 | YOLOv9 [51] | **49.6** | **76.5** | 60.9 | 47.4 | 7 | 10 min |

**Ablation of Region Proposal Methods.** Regional distillation and contrastive learning are highly sensitive to the quality of region proposals. To assess their impact on the performance of FineCLIP, we evaluate four different region proposal methods: manual annotations from COCO, FastSAM [69], RPN [43], and YOLOv9 [51]. The corresponding results are presented in Table 2. These findings highlight three key insights: **1)** Automated region proposals perform comparably to manually annotated high-quality regions. **2)** More region proposals do not necessarily improve performance. Although FastSAM generates the most proposals, but they appear overly cluttered upon manual inspection, resulting in suboptimal model performance. **3)** RPN provides a moderate number of region proposals with satisfactory and balanced performance. YOLOv9, which focuses on specific object categories, produces fewer but more precise proposals, leading to the best box classification performance of FineCLIP.

**Ablation of Region Annotation Methods.** We also explore the impact of region annotation methods on FineCLIP's performance. In addition to manual annotation, two main approaches are used for generating region textual descriptions. The first is a rule-based method [70], which selects the region concept from a predefined concept pool based on similarity scores and integrates it into a description template. The second strategy leverages LVLM for region annotation.

Table 3: Performance comparisons of FineCLIP using different region annotation methods.

| # | Region Annotation Method | Box Classification | | Retrieval | |
|---|---|---|---|---|---|
| | | Top1 | Top5 | I2T | T2I |
| 1 | Rule-base [70] | 43.1 | 71.3 | 58.6 | 46.9 |
| 2 | Intern-XComposer [67] | 47.0 | **75.9** | 60.1 | 45.1 |
| 3 | BLIP2 [26] | **48.4** | 75.6 | **62.2** | **47.6** |

Specifically, we annotate the boxes in COCO `train2017` split using the rule-based method, BLIP-2 [26], and InternLM-XComposer [67], and evaluate their impact on FineCLIP's performance. For LVLMs, we use the prompt: "Describe this image in one sentence." As shown in Table 3, LVLMs outperforms rule-based method, highlighting the effectiveness of LVLMs in generating valuable fine-grained knowledge. Notably, BLIP-2 provides the greatest improvements to FineCLIP.

Table 4: Performance comparisons of FineCLIP and competing methods on COCO.

| # | Methods | Box Classification | | Retrieval | | Time Overhead (per epoch) | GPU Memory Usage (per card) |
|---|---|---|---|---|---|---|---|
| | | Top1 | Top5 | I2T | T2I | | |
| 1 | Pre-trained CLIP [40] | 31.1 | 53.7 | 59.3 | 42.4 | - | - |
| 2 | CLIP [40] | 42.3 | 66.6 | 62.4 | 48.8 | 6 min | 8G |
| 3 | RegionCLIP [70] | 40.0 | 65.3 | 25.1 | 31.2 | 9 min | 5G |
| 4 | CLIPSelf [55] | 43.7 | 72.3 | 33.3 | 21.2 | 10 min | 6G |
| 5 | FineCLIP(Ours) | 48.4 | 75.6 | 62.2 | 47.6 | 11 min | 36G |

## 3.2 Comparisons with Competing Methods

Following the evaluation setting in Subsection 3.1, we compare FineCLIP with three closely related approaches: CLIP [40], RegionCLIP [70], and CLIPSelf [55]. To ensure a fair comparison, all methods adopt the ViT-B/16 as backbone and input images of $224 \times 224$ resolution. As presented in Table 4, FineCLIP demonstrates the most improvement over Pre-trained CLIP in both dense feature extraction and global representation. In contrast, while RegionCLIP and CLIPSelf achieve moderate gains in box classification tasks, they struggle to maintain the important visual-semantic consistency.

Additionally, we report the time overhead and GPU memory usage of the competing methods during training on COCO `train2017` split. Due to FineCLIP's incorporating of the multi-grained contrastive learning paradigm and the self-distillation scheme, it requires comparatively higher GPU memory usage. Nevertheless, it's worth highlighting that the per-epoch training time for FineCLIP (11 minutes) is only marginally longer than that of CLIPSelf (10 minutes) and RegionCLIP (9 minutes). Therefore, the training time for FineCLIP remains well within acceptable limits.

## 3.3 Comparisons on Scaled Trainset

**Data Preparation.** In this subsection, we evaluate the performance of FineCLIP trained on a scaled dataset. We begin by constructing the trainset based on the Conceptual Caption dataset (CC3M) [47], which comprises 3 million image-text pairs sourced from the internet. To meet training data requirements of FineCLIP, we follow a three-step process to create region-text pairs.

*Image Filtering*: This step retains images with rich contents to facilitate the acquisition of clear and valuable regional proposals. Specifically, we filter out low-resolution images and those that fail to generate regions via the region proposal model. After this process, we retain 2.5 million high-resolution images from CC3M, referred to *"CC2.5M"*.

*Region Proposal*: Based on ablation results in Table 2, we select YOLOv9 [51] to detect objects in images. This process yields 10.4 million high-quality regions (approximately four regions per image) and takes around 4.5 hours to complete.

*Region Annotation*: According to results shown in Table 3, we utilize the BLIP2-COCO-6.7B model to annotate region proposals, which takes approximately 12.5 hours.

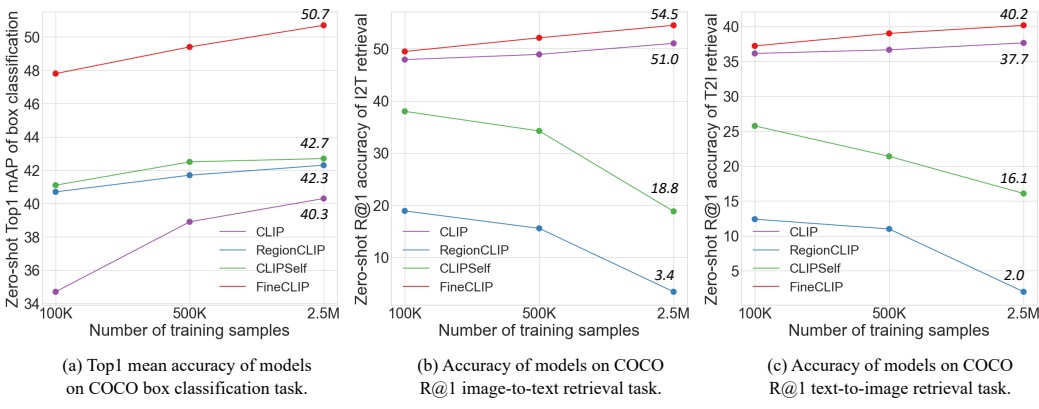

(a) Top1 mean accuracy of models on COCO box classification task.

(b) Accuracy of models on COCO R@1 image-to-text retrieval task.

(c) Accuracy of models on COCO R@1 text-to-image retrieval task.

Figure 2: Zero-shot comparisons of models pre-trained on datasets in three different scales.

**Zero-shot Comparisons.** To investigate the impact of data scale on model performance, we sample three trainsets of varying sizes from CC2.5M: 100K, 500K, and 2.5M samples. We train FineCLIP and competing methods, including CLIP [40], RegionCLIP [70], and CLIPSelf [55], using the official open-source code on these three trainsets. To ensure fairness, all methods involved in comparisons adopt ViT-B/16 as backbone and input images of $224 \times 224$ resolution. For each dataset, we train these models for 6 epochs and then evaluate their zero-shot performance on COCO benchmark, using the same metrics of ablation study in Section 3.1.

In Figure 2, we present the accuracy curves of four methods on three tasks as the dataset scales up, with detailed quantitative results shown in Appendix Table 11. Overall, our FineCLIP outperforms the other three competing methods in all cases. Specifically, in terms of fine-grained understanding, as shown in Figure 2(a), FineCLIP significantly surpasses other methods in the box classification task, with a remarkable +10.4 mAP over CLIP and +8.0 mAP over CLIPSelf. Notably, as the dataset size increases, FineCLIP's performance continues to grow rapidly, showing promising scalability, whereas the growth rates of RegionCLIP and CLIPSelf slow down, indicating that their training gradually converges to a relatively low performance level.

As for the evaluation of image-text alignment, according to Figure 2(b) and (c), FineCLIP even surpasses CLIP in retrieval tasks, demonstrating that enhanced local representation and the acquisition of fine-grained knowledge contribute to more robust global embeddings. In contrast, RegionCLIP and CLIPSelf fail to maintain semantic consistency in visual embeddings, with their performance deteriorating as the trainset size increases.

**Visualization Results.** Figure 3 presents the visualized attention maps of our FineCLIP on images responding to complete sentences or individual words. We can see from Figure 3(a)-(c) that FineCLIP comprehends sentence semantics and identifies related elements, even tiny objects like a "man" or irregularly shaped items such as a "kite." Figure 3(d)-(e) shows that FineCLIP can well locate different objects within the same image. Interestingly, FineCLIP can also capture abstract concepts such as "looking into the distance" in Figure 3(c)) and actions. For instance, when recognizing "riding", FineCLIP focuses on both the rider and the horse, while for "watching soccer", it highlights human faces and the soccer ball on the ground. These results collectively indicate that FineCLIP effectively learns to understand fine-grained semantics.

## 3.4 Application to Fine-grained Localization

**Open-Vocabulary Object Detection.** To evaluate whether the improved fine-grained understanding learned with FineCLIP translates to tasks requiring fine-grained localization, we serve FineCLIP as the backbone for open-vocabulary object detection. Following the previous work [55], we build open-vocabulary object detectors based on F-ViT architecture, which is a two-stage detector baseline built on frozen CLIP ViTs. Considering that the input resolution has a significant influence on detection performance, to ensure the comparison fairness, we utilize the checkpoints of FineCLIP and competing methods trained on CC2.5M with input image size of $224 \times 224$ for ViT-B/16 and $336 \times 336$ for ViT-L/14 to initialize the F-ViT. In training hyper-parameters, we employ AdamW

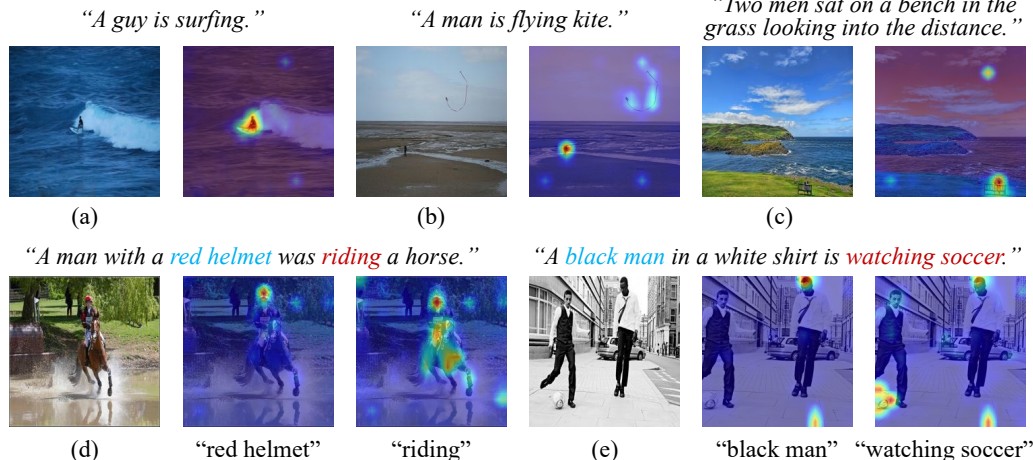

*"A guy is surfing."*      *"A man is flying kite."*      *"Two men sat on a bench in the grass looking into the distance."*

(a)      (b)      (c)

*"A man with a red helmet was riding a horse."*      *"A black man in a white shirt is watching soccer."*

(d)     "red helmet"     "riding"      (e)     "black man"    "watching soccer"

Figure 3: Visualizations of attention maps of our FineCLIP using GAE [5] on images responding to complete sentences or individual words. (a)-(c) Image attention maps w.r.t. different sentences. (d)(e) Image attention maps w.r.t. different words.

Table 5: Results on open-vocabulary object detection. † means the CLIP ViT backbone is initialized with the checkpoint of the corresponding method trained on CC2.5M.

(a) OV-COCO benchmark

| Method | Backbone | $AP_{50}^{novel}$ | $AP_{50}^{base}$ | $AP_{50}$ |
|---|---|---|---|---|
| OV-RCNN [63] | RN50 | 17.5 | 41.0 | 34.9 |
| RegionCLIP [70] | RN50 | 26.8 | 54.8 | 47.5 |
| PB-OVD [11] | RN50 | 30.8 | 46.1 | 42.1 |
| Detic [73] | RN50 | 27.8 | 51.1 | 45.0 |
| VLDet [29] | RN50 | 32.0 | 50.6 | 45.8 |
| F-VLM [23] | RN50 | 28.0 | - | 39.6 |
| BARON-Cap [54] | RN50 | 33.1 | 54.8 | 49.1 |
| CORA [56] | RN50 | 35.1 | 35.5 | 35.4 |
| RO-ViT [20] | ViT-B/16 | 30.2 | - | 41.5 |
| RO-ViT [20] | ViT-L/16 | 33.0 | - | 47.7 |
| CFM-ViT [19] | ViT-L/16 | 34.1 | - | 46.0 |
| F-ViT | ViT-B/16 | 17.5 | 41.0 | 34.9 |
| F-ViT+CLIPSelf† | ViT-B/16 | 25.4 | 40.9 | 36.8 |
| F-ViT+FineCLIP† | ViT-B/16 | 29.8$_{\uparrow12.3}$ | 45.9$_{\uparrow4.9}$ | 41.7$_{\uparrow6.8}$ |
| F-ViT | ViT-L/14 | 24.7 | 53.6 | 46.0 |
| F-ViT+CLIPSelf† | ViT-L/14 | 38.4 | 54.4 | 50.2 |
| F-ViT+FineCLIP† | ViT-L/14 | 40.0$_{\uparrow15.3}$ | 57.2$_{\uparrow3.6}$ | 52.7$_{\uparrow6.7}$ |

(b) OV-LVIS benchmark

| Method | Backbone | $mAP_c$ | $mAP_f$ | $mAP_r$ | mAP |
|---|---|---|---|---|---|
| F-ViT+CLIPSelf† | ViT-B/16 | 7.6 | 10.5 | 10.6 | 9.3 |
| F-ViT+FineCLIP† | ViT-B/16 | 8.0 | 10.9 | 10.4 | 9.5 |
| F-ViT+CLIPSelf† | ViT-L/14 | 19.2 | 22.0 | **20.6** | 20.5 |
| F-ViT+FineCLIP† | ViT-L/14 | **19.6** | **23.1** | 20.2 | **20.9** |

(c) Transfer evaluation of the LVIS-trained detector on COCO and Object365 benchmarks.

| Method | Benchmark | AP | $AP_{50}$ | $AP_{75}$ | $AP_s$ | $AP_m$ | $AP_l$ |
|---|---|---|---|---|---|---|---|
| F-ViT+CLIPSelf† | COCO | 32.3 | 51.7 | 34.5 | 13.1 | 35.9 | 54.1 |
| F-ViT+FineCLIP† | COCO | **33.6** | **52.7** | **36.1** | **14.3** | **38.5** | **55.1** |
| F-ViT+CLIPSelf† | Object365 | 12.1 | **19.9** | 12.5 | 2.2 | **13.4** | **30.2** |
| F-ViT+FineCLIP† | Object365 | 12.1 | 19.8 | **12.6** | **2.3** | 13.3 | 29.3 |

optimizer with batch size of $8$, learning rate of $1e - 4$, and weight decay of $0.1$. We train the models for 3 epochs on the OV-COCO [6] benchmark and $48$ epochs on the OV-LVIS [13] benchmark.

For evaluation, we follow previous works [55, 70] to report box AP at IoU $0.5$ of base, novel and all categories ($AP_{50}^{novel}$, $AP_{50}^{base}$ and $AP_{50}$) on OV-COCO, and the AP for base, novel and all categories ($mAP_c$, $mAP_f$, $mAP_r$ mAP) on OV-LVIS as comparison indicators. The results are shown in Table 5. By replacing the frozen CLIP ViTs with FineCLIP checkpoints, F-ViT gains significant performance improvements ($24.7$ vs $40.0$ $AP_{50}^{novel}$ on OV-COCO). Notably, our FineCLIP outperforms the existing open-vocabulary object detection methods on the OV-COCO benchmark under fair training settings. Compared with cutting-edge CLIPSelf, our FineCLIP brings better improvements to baseline F-ViT on both benchmarks. Additionally, we also evaluate the detector trained on OV-LVIS on the validation split of COCO and object365 [46] v1 datasets, with results shown in Table 5c. FineCLIP surpasses CLIPSelf on the COCO benchmark and achieves similar performance on the Object365 benchmark. All these results demonstrate the effectiveness of our FineCLIP in fine-grained understanding.

**Open-Vocabulary Semantic Segmentation.** Next, we explore the performance of FineCLIP when applied to the open-vocabulary semantic segmentation task. Following the previous work [55], we

Table 6: Results on open-vocabulary semantic segmentation. † means the CLIP ViT backbone is initialized with the checkpoint of the corresponding method trained on CC2.5M.

| Method | Backbone | ADE-150 | | ADE-847 | | PC-59 | |
|---|---|---|---|---|---|---|---|
| | | mIoU | mAcc | mIoU | mAcc | mIoU | mAcc |
| OVSeg [28] | ViT-B/16 | 24.8 | - | 7.1 | - | 53.3 | - |
| SAN [57] | ViT-B/16 | 27.5 | 45.6 | 10.1 | 21.1 | 53.8 | 73.0 |
| SAN [57] | ViT-L/14 | 32.1 | 50.7 | 12.4 | **25.2** | 57.7 | 77.6 |
| CatSeg [7] | ViT-B/16 | 27.2 | 41.2 | 8.4 | 16.6 | 57.5 | 74.0 |
| CatSeg [7] | ViT-L/14 | 31.5 | 46.2 | 10.8 | 20.5 | **62.0** | **78.3** |
| CatSeg+CLIPSelf† [55] | ViT-B/16 | 29.7 | 45.1 | 10.1 | 17.2 | 55.3 | 73.4 |
| CatSeg+CLIPSelf† [55] | ViT-L/14 | 34.9 | 52.9 | 13.6 | 23.0 | 59.1 | 77.1 |
| CatSeg+FineCLIP† | ViT-B/16 | $32.4_{\uparrow5.2}$ | $50.5_{\uparrow9.3}$ | $12.2_{\uparrow4.2}$ | $22.2_{\uparrow5.6}$ | $56.0_{\downarrow1.5}$ | $74.4_{\uparrow0.4}$ |
| CatSeg+FineCLIP† | ViT-L/14 | $\mathbf{36.1}_{\uparrow4.6}$ | $\mathbf{53.5}_{\uparrow7.3}$ | $\mathbf{14.1}_{\uparrow3.3}$ | $23.8_{\uparrow3.3}$ | $59.9_{\downarrow2.1}$ | $78.3_{\uparrow0}$ |

build the segmentation model based on CatSeg [7], which utilizes the visual dense features of CLIP ViTs (ViT-B/16 and ViT-L/14 from OpenAI) with a cost-aggregation module. To ensure comparison fairness, we train FineCLIP and CLIPSelf models, which are initialized with pre-trained OpenAI CLIP [17], on CC2.5M with the same input image resolution of $384 \times 384$ for ViT-B/16 and $336 \times 336$ for ViT-L/14, and then replace the backbone of CatSeg with FineCLIP or CLIPSelf ViT for the following segmentation fine-tuning. We fine-tune segmentation models on COCO Stuff [4] and evaluate them on ADE20K [71] and PASCAL Context [34] dataset using mean IoU (mIoU) and mean pixel accuracy (mAcc).

As shown in Table 6, FineCLIP brings non-trivial improvements to CatSeg across most evaluation indicators, surpassing the enhancements provided by CLIPSelf. We observe that both FineCLIP and CLIPSelf cause a decrease in PC-59 mIoU, which may be attributed to the data distribution gap between CC2.5M and PASCAL Context. Furthermore, FineCLIP comprehensively improve the mAcc performance of CatSeg, indicating the enhanced per-pixel classification performance.

Table 7: Comparative results on zero-shot image-text retrieval on the Flickr30k and MSCOCO datasets. R@i denotes Recall at i. All approaches adopt ViT-B/16 architecture with input image size of $224 \times 224$. † indicates that the method is initialized with pre-trained CLIP and further trained on CC2.5M. The methods with gray background are pre-trained on large-scale dataset.

| | Flickr30k | | | | | | MSCOCO | | | | | |
|---|---|---|---|---|---|---|---|---|---|---|---|---|
| | image-to-text | | | text-to-image | | | image-to-text | | | text-to-image | | |
| Methods | R@1 | R@5 | R@10 | R@1 | R@5 | R@10 | R@1 | R@5 | R@10 | R@1 | R@5 | R@10 |
| CLIP[40] | 84.0 | 96.1 | 98.2 | 71.6 | 90.3 | 94.1 | 56.2 | 80.6 | 88.2 | 42.4 | 68.6 | 78.3 |
| SPARC[3] | 84.4 | 97.6 | 98.7 | 72.0 | 91.2 | 94.9 | 57.6 | 81.2 | 88.5 | 43.0 | 68.6 | 78.5 |
| PACL[35] | 69.6 | 89.7 | 94.2 | 54.9 | 80.7 | 87.3 | 41.8 | 67.8 | 77.6 | 29.1 | 54.3 | 65.5 |
| GLoRIA[16] | 78.0 | 95.5 | 98.0 | 68.4 | 88.9 | 93.2 | 49.7 | 75.4 | 84.6 | 38.9 | 65.1 | 75.2 |
| MGCA[52] | 82.2 | 96.1 | 98.1 | 67.7 | 88.5 | 93.2 | 57.6 | 80.5 | 87.8 | 39.8 | 65.7 | 75.3 |
| FILIP[58] | 69.0 | 89.8 | 94.0 | 55.8 | 81.5 | 87.9 | 40.2 | 66.0 | 76.3 | 29.5 | 55.3 | 66.3 |
| CLIP† [40] | 81.6 | 96.2 | 98.0 | 64.9 | 88.3 | 93.6 | 51.1 | 76.4 | 84.9 | 37.6 | 63.9 | 74.3 |
| RegionCLIP†[70] | 3.9 | 12.2 | 18.4 | 7.9 | 22.7 | 71.3 | 2.0 | 7.1 | 11.5 | 3.4 | 11.8 | 19.0 |
| CLIPSelf†[55] | 33.8 | 61.7 | 73.0 | 35.0 | 61.3 | 32.7 | 18.8 | 38.9 | 50.4 | 16.1 | 34.5 | 45.1 |
| FineCLIP† | **82.5** | **96.4** | **98.6** | **67.9** | **89.1** | **94.1** | **54.5** | **78.6** | **85.8** | **40.2** | **66.5** | **76.1** |

## 3.5 Application to Image-level Task

**Zero-shot Image-Text Retrieval.** We further evaluate FineCLIP on zero-shot cross-modal retrieval tasks using Flicker30K [39] and MSCOCO [30], with the results presented in Table 7. Previous works, such as SPARC [3], PACL [35], GLoRIA [16], MGCA [52], and FILIP [58], introduce fine-grained losses to extract token-level cross-modal alignments. The results of these methods, re-implemented using the same pre-training datasets (approximately 3.2 billion data points), architecture, and training steps, are taken from SPARC [3]. For fair comparisons, we train competing methods, initialized with pre-trained CLIP parameters, on CC2.5M for 4 epochs.

Given that CC2.5M has in a limited data distribution, further training of pre-trained models on this dataset inevitably leads to more or less performance decay. Compared to the baseline CLIP, FineCLIP better maintains retrieval performance, demonstrating its effectiveness in enhancing global

embeddings and capturing valuable semantics. In contrast, RegionCLIP and CLIPSelf struggle to maintain global embeddings for addressing retrieval tasks. Surprisingly, FineCLIP even outperforms most pre-trained models with fine-grained losses, strongly supporting the effectiveness of FineCLIP.

## 4 Related Work

### 4.1 Fine-grained Understanding in Vision-language Models

Early works focused on learning from intensive human labels by training image classifiers [9, 14, 22, 48, 50]. These classifiers focus only on a limited range of objects, making it difficult to cover a wide range of semantics. CLIP [40] and its diverse variants [12, 18, 68] popularized learning general visual-language representations by pre-training on noisy large-scale datasets like LAION-400M [44] and LAION-5B [45], which exhibit potent representation capabilities and exceptional generalizability.

Despite the great achievements, CLIP model has shown weak alignment between regions and corresponding texts [37, 41, 62]. This problem can be roughly attributes to limitations of 1) CLIP loss which ignores the supervision of visual and textual dense features, and 2) brief coarse captions that are insufficient to allow the model in comprehending image details [10]. Recent works attempted to enhance CLIP's fine-grained understanding by building strong region-text alignment. One line of work leverages region-level annotations for vision-language pre-training [59, 64, 66]. For instance, GLIP [27] utilized large-scale human-labeled grounding data to align semantics at phase and region level, which achieved stronger performance on fully-supervised detection benchmarks. Region-CLIP [70] proposed to generate region descriptions by filling plausible concepts into pre-defined templates. Since the semantics of synthesized descriptions are limited by the pre-defined concept pool, RegionCLIP essentially models the finite category classification task. Although this approach brings benefits to downstream detection tasks, it still struggles to cover broad semantic diversity of open-world scenarios. Another remarkable work is CLIPSelf [55], which proposed to distill the global representation capability of the frozen teacher model to dense feature extraction of the student model. While CLIPSelf successfully enhancing the local representations, the frozen teacher model limits the performance ceiling of the student model, which is not consistent with the intention of pre-training. A separate line of work explored incorporating losses between image patch and text token embeddings to learn representations encoding more fine-grained details [3, 16, 35, 52, 60].

### 4.2 Fine-grained Image Annotation

The process of generating region-text pairs from images involves two steps: proposing regions and annotating the regions with texts. Common strategies for region proposal include random cropping, using Region Proposal Network (RPN) [43] or detectors [42, 65] to generate bounding boxes around objects, or leveraging segmentation models [24] such as SAM [21]. Region annotations can be obtained by expert manual labeling, synthesizing captions using traditional NLP techniques, or generating them with models. A notable work of the second way is Kosmos-2 [38], which introduced a pseudo-labeling pipeline that utilizes the pre-trained GLIP [27] to automatically generate fine-grained pseudo-labels of region boxes. In the era of large models, recent LVLMs [2, 31, 36, 53] have demonstrated impressive capabilities of visual understanding, instruction-following and generalization. By setting prompts, users can control the characteristics of outputs, such as length and writing style, to obtain high-quality responses, making it suitable for image fine-grained annotation.

## 5 Conclusion

In this work, we present FineCLIP, a coherent framework that unifies cross-model multi-grained alignment and uni-modal global-to-region guidance. FineCLIP effectively leverages diverse semantics from automatically generated regional data and enhances the interactions of multi-grained complementary information through a real-time self-distillation scheme. Extensive experiments demonstrate the superior performance of FineCLIP in fine-grained understanding tasks, including box classification, open-vocabulary object detection and segmentation, as well as in global representation tasks like image-text retrieval. We believe this study provides valuable insights into related fields.

## Acknowledge

This work is partially supported by National Natural Science Foundation of China (62376274, 62437002) and Beijing Natural Science Foundation (L233008).

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

# A Appendix

## A.1 More Ablation Study

**Ablation of Input Image Sizes.** To explore the impact of input image size on FineCLIP, we set up four different image resolutions [224, 320, 448, 640] used for training and inference. To fit different image resolutions, the position encoding of ViT is up-sampled to the target shape through bicubic interpolation. According to the results shown in Table 8, as we gradually increase the image resolution from 224 to 640, FineCLIP's performance on box classification

Table 8: Ablation study on input image sizes.

| # | Image Size | Box Classification | | Retrieval | |
|---|---|---|---|---|---|
| | | Top1 | Top5 | I2T | T2I |
| 1 | 224 | 48.4 | 75.6 | 62.2 | 47.6 |
| 2 | 320 | 51.2 | 78.0 | 63.3 | 48.0 |
| 3 | 448 | 52.6 | 79.7 | 62.9 | 48.2 |
| 4 | 640 | 54.3 | 80.6 | 60.7 | 46.9 |

task improves due to the increasingly detailed information provided by images. On the other hand, when the image size becomes too large, ViT receives an excessive number of image patches, which greatly increases the complexity for the global embedding to summarize global information, leading to a rise and then a decline in FineCLIP's performance on retrieval tasks as the image size increases.

**Ablation of ViT Backbones.** We also explores the influence of using different ViT backbones on FineCLIP. As shown in Table 9, when FineCLIP employ ViT-L/14, which has large number of parameters, as the initialization, FineCLIP achieves significantly improvements on both types of tasks compared to using ViT-B/16. This result demonstrates the general applicability of our method to different ViT architectures.

Table 9: Ablation study on ViT backbones.

| # | Backbone | Params | Box Classification | | Retrieval | |
|---|---|---|---|---|---|---|
| | | | Top1 | Top5 | I2T | T2I |
| 1 | ViT-B/16 | 149M | 48.4 | 75.6 | 62.2 | 47.6 |
| 2 | ViT-L/14 | 428M | 52.6 | 79.2 | 66.0 | 53.4 |

Table 10: The study on the impact of $L_{SD}$ to FineCLIP performance in zero-shot setting across different amount of training samples.

| Num of Training Samples | | 100K | | | | 500K | | | | 2.5M | | | |
|---|---|---|---|---|---|---|---|---|---|---|---|---|---|
| # | Objective Function | Box Classification | | Retrieval | | Box Classification | | Retrieval | | Box Classification | | Retrieval | |
| | | Top1 | Top5 | I2T | T2I | Top1 | Top5 | I2T | T2I | Top1 | Top5 | I2T | T2I |
| 1 | $L_{GC} + L_{RC}$ | 46.6 | 71.2 | 50.6 | 38.4 | 48.2 | 73.1 | 51.6 | 39.0 | 49.5 | 74.7 | 54.3 | 39.8 |
| 2 | $L_{GC} + L_{RC} + L_{SD}$ | 47.8 | 74.1 | 49.5 | 37.2 | 49.4 | 89.7 | 51.1 | 39.0 | 50.7 | 91.4 | 54.4 | 40.2 |

**Effect Exploration of $L_{SD}$ on Zero-shot Setting.** We conduct further validation under zero-shot setting by using trainset in different scales, with results shown in Table 10. We observe an interesting trend: as the scale of trainset increases, the negative impact of $L_{SD}$ on retrieval tasks gradually diminishes. This means that $L_{SD}$ tends to have a positive impact on retrieval performance when FineCLIP is fine-tuned with larger-scale data.

Table 11: Results of zero-shot comparisons with datasets in different scales.

| Num of Training Samples | | 100K | | | | 500K | | | | 2.5M | | | |
|---|---|---|---|---|---|---|---|---|---|---|---|---|---|
| # | Method | Box Classification | | Retrieval | | Box Classification | | Retrieval | | Box Classification | | Retrieval | |
| | | Top1 | Top5 | I2T | T2I | Top1 | Top5 | I2T | T2I | Top1 | Top5 | I2T | T2I |
| 1 | CLIP | 34.7 | 69.7 | 47.9 | 36.1 | 38.9 | 73.2 | 48.9 | 36.7 | 40.3 | 75.6 | 51.0 | 37.7 |
| 2 | RegionCLIP | 40.7 | 65.6 | 18.9 | 12.4 | 41.7 | 65.7 | 15.6 | 11.0 | 42.3 | 63.4 | 3.4 | 2.0 |
| 3 | CLIPSelf | 41.1 | 69.7 | 38.0 | 25.8 | 42.5 | 70.7 | 34.2 | 21.4 | 42.7 | 70.9 | 18.8 | 16.1 |
| 4 | FineCLIP (Ours) | **47.8** | **74.1** | **49.5** | **37.2** | **49.4** | **89.7** | **51.1** | **39.0** | **50.7** | **91.4** | **54.4** | **40.2** |

## A.2 Detailed Zero-shot Comparison Results

We provide the detailed zero-shot comparison results on COCO dataset in Table 11. FineCLIP significantly outperforms the other three comparison algorithms in all scenarios and maintains the fastest performance growth as the dataset expands. Surprisingly, FineCLIP's performance in the retrieval tasks under the zero-shot setting even surpasses CLIP, indicating that FineCLIP can learn valuable regional semantics, and further enhance its global understanding capability.

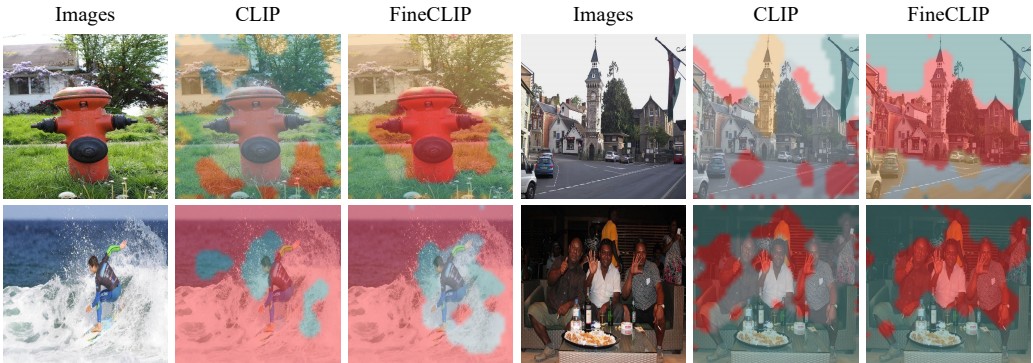

Figure 4: K-Means visualization of the dense features of CLIP ViT. We show the raw images, the K-means results of the pre-trained CLIP ViT, and those of our FineCLIP ViT.

Table 12: Comparisons on OV-COCO benchmark with CLIPSelf training settings.

| Method | Backbone | Region Type | Input Image Size | $AP_{50}^{novel}$ | $AP_{50}^{base}$ | $AP_{50}$ |
|---|---|---|---|---|---|---|
| F-ViT | ViT-B/16 | - | - | 17.5 | 41.0 | 34.9 |
| F-ViT+CLIPSelf | ViT-B/16 | Region Proposal | 1024 | **37.6** | 54.9 | 50.4 |
| F-ViT+FineCLIP | ViT-B/16 | Region Proposal | 640 | 33.5 | **57.8** | **51.4** |

### A.3 Comparisons with CLIPSelf Training Setting

We also evaluate the performance of FineCLIP in open-vocabulary detection tasks when adopting settings similar to CLIPSelf. Following CLIPSelf, we train FineCLIP on COCO Train2017 split, using the region proposals from the trainset and region captions generated by BLIP2. After fine-tuning, FineCLIP's ViT parameters are used to initialize the F-ViT for downstream OV-COCO training. As FineCLIP is essentially a contrastive learning method, it needs a larger batch size than distillation methods for effective training. Due to the GPU memory limitations, we could only increase the training image resolution of FineCLIP to 640, which is much lower than 1024 used by CLIPSelf.

The results in Table 12 show that despite FineCLIP's lower input image resolution, F-ViT+FineCLIP still outperforms F-ViT+CLIPSelf in $AP_{50}^{base}$(57.8 vs. 54.9) and $AP_{50}$(51.4 vs. 50.4) metrics.

### A.4 More Visualization Results

**K-Means Visualization of Dense Features.** We present K-Means visualization of dense features generated by CLIP and FineCLIP ViT in Figure 4. It can be seen that the dense features produced by CLIP are noisier and exhibit significant instability. In contrast, the results corresponding to FineCLIP are clearer and more consistent with the local semantic.

### A.5 Discussion

In this subsection, we discuss the differences between FineCLIP and two closely related works. We depict the main architectures of CLIP, RegionCLIP, CLIPSelf and FineCLIP in Figure 5.

**Relation to RegionCLIP.** RegionCLIP proposes a regional contrastive learning paradigm based on synthesized region concepts for enhancing fine-grained understanding. However, these synthesized region concepts, generated by filling pseudo object label selected from a label pool into a pre-defined description template, lack sufficient semantic diversity. Considering that the negative textual samples for contrastive are object concepts that are not matched to the region but matched to other regions in the batch, it essentially models an object classification tasks with predefined finite categories. In contrast, our FineCLIP learns from diverse and detailed regional descriptions generated by LVLM, which are more representative of real-world scenarios.

**Relation to CLIPSelf.** CLIPSelf builds a promising uni-modal global-to-local guidance scheme, relying on a pre-trained frozen teacher model. We think the main advantage of this scheme is the

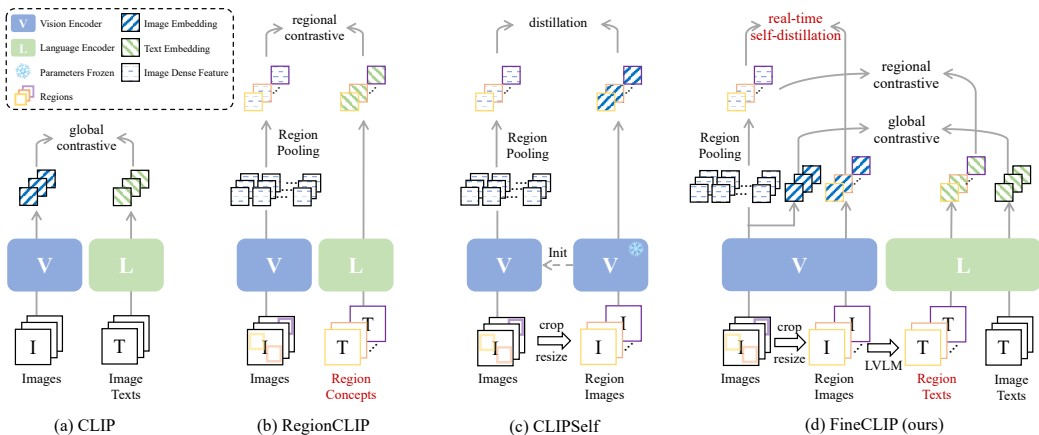

Figure 5: Illustration of CLIP variants.

introduction of *position inductive bias*, enabling dense features to grasp local semantics around them, which is crucial for most downstream tasks. However, a notable shortcoming of CLIPSelf is that the frozen teacher model limits the performance of student model, and also discrpts the semantic consistency in visual features. By comparison, our FineCLIP fully leverages the model to teach itself with the help of global contrastive learning. While retaining this position inductive bias, FineCLIP enhances local features with real-time global embeddings enhanced by contrastive learning, thereby achieving more valuable local representations.

## A.6 Broader Impact

Our FineCLIP contributes to develop a coherent learning framework that enhances CLIP's fine-grained understanding capabilities. With the growing adoption of transformers, FineCLIP is expected to better address various downstream tasks. To ensure a positive social impact, we conduct experiments using academic open-source datasets that do not involve personal privacy issues.

## A.7 Limitation

FineCLIP has strict region-level data requirements. Additionally, due to the limitations in computational resources, we cannot afford to train FineCLIP from scratch on the billion-level dataset, making it difficult to fully realize its potential.

## A.8 Future Direction

In this work, we address fine-grained vision-language representation learning by introducing generated region-text pairs and developing a unified architecture. While FineCLIP has shown initial success, there is still significant room for improvement in region-text generation.

Firstly, existing region proposal methods struggle to balance category richness and accurate segmentation. Our experiments with existing RPN, detectors, and segmentation models revealed that they either limit the number of categories or produce disorganized proposals. This results in suboptimal performance on tasks with numerous categories, such as the LVIS benchmark. One potential improvement is to train a more robust RPN on datasets with a greater variety of categories, enhancing region proposal quality. Secondly, due to computational constraints, we did not use the most powerful LVLM for annotations. A stronger LVLM would intuitively produce higher quality annotations.

We hope these insights inspire future research.

