# OpenReview forum: "FineCLIP: Self-distilled Region-based CLIP for Better Fine-grained Understanding"
_NeurIPS.cc/2024/Conference — NeurIPS 2024 poster_

### Official Review · Reviewer_rzQ8 · 2024-07-07

**Soundness:** 2
**Presentation:** 3
**Contribution:** 3
**Rating:** 5
**Confidence:** 5

**Summary:**

This paper integrates three existing techniques in vision-language pre-training into a single end-to-end fine-tuning framework, referred to as FineCLIP:
- global contrastive: This aligns the global representation of an image with the text embedding.
- reginal contrastive: This aligns the pooled region-level representations, defined by the generated proposals, with the regional caption generated by a LVLM.
- regional self-distillation: This aligns the pooled region-level representations with the global representations of the corresponding local crops, following the approach of CLIPSelf [1]. Uniquely, it employs the updated vision encoder itself as the teacher.

In addition, the authors construct a new subset, “CC2.5M”, based on CC3M for further fine-tuning.

[1] CLIPSelf: Vision Transformer Distills Itself for Open-Vocabulary Dense Prediction. ICLR 2024

**Strengths:**

- Compared to its close competitors, CLIPSelf and RegionCLIP, the proposed method demonstrates a performance gain on various dense-level tasks while preserving the image-level inference capability.
- The method integrates the pre-training strategies of both CLIPSelf and RegionCLIP into a single unified framework. This could potentially benefit the pre-training of vision-language foundational models, facilitating both coarse-grained and fine-grained understanding.

**Weaknesses:**

__W1: limited practical impact.__

- As a direct successor to CLIPSelf, the performance on both OV-COCO and OV-LVIS benchmarks is significantly lower than the original baseline of CLIPSelf. For instance, FineCLIP achieves 29.8 AP$_{50}^{novel}$ on OV-COCO with ViT-B/16, while CLIPSelf reaches 37.6 with an even smaller dataset. I understand that they’re under different input resolutions, but why not directly increase the input resolution of FineCLIP instead of increasing the scale of the dataset?

- Moreover, a significant advantage of FineCLIP is its ability to preserve the power of image-level inference. However, according to Table 5, all the fine-tuning strategies cause a performance drop to some extent compared to the original pre-trained CLIP on all the image-level benchmarks.

__W2: The experiments are insufficient and incomplete.__
- Input resolution: Input resolution: Since CLIPSelf can consistently benefit from higher image resolution, a comparison with CLIPSelf under a relatively low resolution (224$\times$224) is unfair. I would suggest including the performance comparison with higher resolution under the original settings of CLIPSelf.

- The mean mask AP on rare categories (mAP$_r$) is missing for OV-LVIS.

__W3:The limitation of CLIPSelf stated in this paper remains unsolved.__
> line150: performance is capped by the teacher’s capabilities.

The supervision of self-distillation heavily relies on the quality of the image-level representation of the teacher (or the updated visual encoder of FineCLIP), but according to Table 5, the image-level performance suffers from a significant drop when compared to the frozen encoder. It’s somewhat unconvincing to me to draw the conclusion that the updated visual encoder can benefit the self-distillation process.

Perhaps the proposed strategy could potentially be beneficial to large-scale vision-language pre-training from scratch given the observations of fine-tuning on CC2.5M, but the current results of this paper are not very convincing to me. I am open to reconsidering my score if my concerns here are addressed.

**Questions:**

- Most of my questions are outlined in the weaknesses section.

- Minor: Some detailed settings are missing. For instance, for Cat-Seg+CLIPSelf and Cat-Seg+FineCLIP, is the region defined by the generated proposals or patches (grids)?

**Limitations:**

Limitations are properly discussed.

---

> ### Author Rebuttal · Authors · 2024-08-07
>
> Q1: The performance of FineCLIP on both OV-COCO and OV-LVIS benchmarks is significantly lower than the original baseline of CLIPSelf. Why not directly increase the input resolution of FineCLIP instead of increasing the scale of the dataset?
>
>
> A1: Thanks for your suggestion. We appreciate the opportunity to clarify the performance of FineCLIP in open-vocabulary detection tasks when adopting settings similar to CLIPSelf.
>
>
> Following CLIPSelf, we train FineCLIP on COCO Train2017 split, using the region proposals from the trainset and region captions generated by BLIP2.
> After fine-tuning, FineCLIP's ViT parameters are used to initialize the F-ViT for downstream OV-COCO training.
> As FineCLIP is essentially a contrastive learning method, it needs a larger batch size than distillation methods for effective training.
> Due to the GPU memory limitations, we could only increase the training image resolution of FineCLIP to 640, which is much lower than 1024 used by CLIPSelf.
>
>
> The results in Table 6 of the attached PDF show that despite FineCLIP's lower training image resolution, F-ViT+FineCLIP still outperforms F-ViT+CLIPSelf in $AP_{50}^{base}$ (57.8 vs. 54.9) and $AP_{50}$ (51.4 vs. 50.4) metrics.
> Due to the time limitation, we could only make this validation on OV-COCO benchmark.
>
> Moreover, we believe that the settings for downstream tasks in our paper are adequately impartial. Even if the scores of different methods might be lower than those reported in original papers, these scores remain fair and reliable for comparison.
> We understand your concern that the current settings may not fully exploit the potential of distillation methods.
> However, we also have to avoid the risk that  high training resolution adversely affects the training stability and effectiveness of contrastive learning methods. In this paper, we thus choose to increase the scale of the dataset instead of increasing the input resolution of FineCLIP.
>
>
> Q2: According to Table 5, all the fine-tuning strategies cause a performance drop to some extent compared to the original pre-trained CLIP on all the image-level benchmarks, which cannot demonstrate FineCLIP's ability to preserve the power of image-level inference.
>
> A2: Good question. We believe the performance decline on image-level benchmarks is caused by both training data and method factors. Therefore, we cannot ignore the impact of the training data, nor can we claim that the method is ineffective based sorely on the performance decline. This is crucial for us to determine whether FineCLIP has a positive or negative effect on image-level task. Considering that different methods used the same training data during fine-tuning, comparing the performance of methods after fine-tuning is sufficient to determine which method is better method. According to Table 5, FineCLIP comprehensively outperforms CLIP after fine-tuning, indicating that FineCLIP maintains stronger global representation capabilities than CLIP. Therefore, we can also reasonably infer that FineCLIP has the potential to exhibit better image-level representation capabilities after pre-training compared to pre-trained CLIP.
>
>
> Q3: Considering that the supervision of self-distillation heavily relies on the quality of the image-level representation of the updated visual encoder of FineCLIP, the image-level performance suffers from a significant drop when compared to the frozen encoder. It's hard to conclude that the updated visual encoder can benefit the self-distillation process.
>
> A3: Good question.
> To begin with, it is important to note that the evaluation on zero-shot retrieval tasks has biases, as it focus more on measuring generalization ability, rather than the representation capability in a specific data domain.
> For instance, as shown in Table 8 in the appendix of our paper, FineCLIP fine-tuned on COCO Train2017 greatly outperforms the pre-trained CLIP.
> Consequently, we believe the image-level performance of FineCLIP's updated encoder is significantly improved in the relevant data domain during the training.
>
>
> A significant issue of the frozen pre-trained encoder is that it cannot learn from training data. In small-scale training scenarios, such as distillation, the frozen pre-trained encoder works well. However, as the training data scale expands, these methods would quickly hit bottlenecks, as indicated by the green line corresponding to CLIPSelf in Figure 2.
> In contrast, the updated visual encoder in our FineCLIP could be consistently improved during the training, thereby better surpporting the self-distillation process.
>
> Furthermore, based on extensive experiments presented in our paper, FineCLIP with an updated encoder surpasses CLIPSelf with a frozen encoder in most evaluation metrics, clearly demonstrating the effectiveness of FineCLIP.
>
>
> Q4: Perhaps the proposed strategy could potentially be beneficial to large-scale vision-language pre-training from scratch given the observations of fine-tuning on CC2.5M, but the current results of this paper are not very convincing to the reviewer.
>
> A4: We believe the distinction between per-training and fine-tuning primarily lies in the scale of training data. Even when FineCLIP is initialized with pre-trained CLIP parameters, its training stage can also be considered as pre-training, if the data scale is sufficiently large. Therefore, there is no fundamental difference between pre-training and fine-tuning. In Section 4.2, by continuously expanding the data scale, Figure 2 shows that FineCLIP demonstrates excellent scalability in both global and local representation capabilities, indicating that our method has great potential to build large-scale vision-language pre-training models.
>
>
> Q5: For Cat-Seg+CLIPSelf and Cat-Seg+FineCLIP, is the region defined by the generated proposals or patches (grids)?
>
> A5: To ensure fairness, all models trained on CC2.5M utilize generated region proposals.

---

> > ### Comment · Reviewer_rzQ8 · 2024-08-11
> > **Replying to Rebuttal by Authors**
> >
> > Thanks for the response. Most of my concerns have been adequately addressed, but I don't fully agree with A4. For FineCLIP, the self-distillation mechanism relies on high-quality supervision from $[CLS]$, which may not be compatible with an initialized visual encoder for pre-training from scratch. Anyway, I will raise my score to 5, as the method appears to be effective for fine-tuning.

---

> > > ### Author Response · Authors · 2024-08-11
> > > **Replying to Reviewer rzQ8**
> > >
> > > We sincerely appreciate your positive feedback and insightful comments. We will carefully incorporate the key points of our discussion into the revision. Thank you very much again.

---

> ### Author Response · Authors · 2024-08-08
> **Further Clarification on Q2 and Q3**
>
> We greatly appreciate your constructive comments and suggestions. We'd like to address your concerns regrading the image-level performance of FineCLIP with further clarification.
>
> It should be noted that Table 5 presents the performance of models on **zero-shot** retrieval benchmarks.
> These zero-shot benchmarks (i.e., Flickr30k and MSCOCO) involve the evaluation data that is somewhat out-of-distribution w.r.t. the trainset.
> As you have pointed out, we had also noticed that the domain gap between the trainset and evaluation data might degrade the model's zero-shot performance when we were preparing this paper submission.
>
> Importantly, as shown in Figure 2, we find that data scale is crucial for zero-shot capabilities, and both CLIP and FineCLIP demonstrate excellent scalability.
> In this paper, we thus chose to use the largest dataset within our capacity, CC2.5M, as the trainset to preserve the model's zero-shot ability as much as possible.
> According to Table 5, the performance drop for CLIP or FineCLIP is not that significant.
> Given the scalability of CLIP and FineCLIP demonstrated in Figure 2, we can reasonably infer that their zero-shot image-level performance could be further improved and even surpasses that of the pre-trained CLIP by increasing the trainset scale.
>
> Moreover, it should also be noted that Table 5 does not show the model's performance on in-domain data.
> According to Table 8 of the Appendix, when models are trained and evaluated on COCO, FineCLIP's performance on retrieval tasks significantly surpasses that of pre-trained CLIP.
> This result indicates that FineCLIP greatly enhances image-level representation on in-domain data, enabling it to better benefit the self-distillation process than pre-trained CLIP.
>
> We hope that the above clarification could address your concerns. We are looking forward to your feedback if there is still any confusion. Thank you very much again.

---

### Official Review · Reviewer_Ask3 · 2024-07-08

**Soundness:** 3
**Presentation:** 4
**Contribution:** 4
**Rating:** 5
**Confidence:** 4

**Summary:**

This paper attempts to overcome the problem of CLIP lacking fine-grained details when adapting to dense prediction tasks. It proposes a unified framework with three training losses: contrastive loss for global image-text pair, region alignment for region-region annotation, and self-distillation for image region embedding and region feature. By combining the three aspects, the FineCLIP shows its effectiveness in fine-grained understanding, at the same time maintains the global representation alignment.

**Strengths:**

1.	The proposed FineCLIP incorporates multi-grained contrastive learnings and a self-distillation scheme, which achieves better fine-grained understanding and also keeps the image-level performance.
2.	This paper is well-written and clearly describes the technical details.
3.	The experiments are extensive (dense prediction tasks and image-level tasks on multiple datasets) and impressive.

**Weaknesses:**

The innovation somewhat combines the previous works regionCLIP and CLIPSelf on finetuning of CLIP, although this work shows evolved implementation details, such as adopting LVLMs to generate region annotations and self-distillation instead of frozen teacher.

**Questions:**

1. I noticed that in Sec.4.1 ablation study, the results in Tab. 1 are obtained by training FineCLIP on the train2017 split of the COCO dataset, while in Sec. 4.2 the training set is CC3M. With the same COCO validation set, training with CC3M shows worse performance on retrieval task (54.5 vs. 62.2, 40.2 vs. 47.6) but better top1 box classification result (50.7 vs. 48.4). Any idea about this result?
2. About region proposal and annotation, I’m curious about some details:
- Are the region information prepared before the CLIP finetuning or real-time processing during the training?
- Will different region proposal methods have a big influence on the final results since it is mentioned that “after careful manual evaluation, we choose YOLOv9” in P6 L237? Maybe you can provide the details about the procedure in the appendix.
- The details about "We use pixel count as the criterion and discard images that fail to propose regions in the following process." to choose CC2.5M.
3. CLIPSelf is an ICLR2024 accepted paper, not CVPR 2024. The reference of [57] is not correct.
4. The table head ("image-to-text" and "text-to-image") of Tab. 5 doesn't align well with the below contents.

---

> ### Author Rebuttal · Authors · 2024-08-07
>
> Q1: With the same COCO validation set, training with CC2.5M shows worse performance on retrieval task but better top1 box classification accuracy. Any idea about this result?
>
> A1: Good question. We believe this result can be explained by the differences between retrieval and box classification tasks, as well as the effectiveness of FineCLIP in utilizing region-text pairs to enhance local feature extraction capabilities.
>
> i) Task Complexity Differences. From the perspective of text descriptions alone, there is a significant difference in class complexity between retrieval and box classification tasks. The retrieval task involves diverse global text descriptions that are flexible combinations of many objects, making it highly sensitive to the overall image content and corresponding textual content and style. This data distribution gap for retrieval task between trainset and testset is hard to eliminate. In contrast, the box classification task focuses on identifying a relatively smaller number of local objects with clearer concepts. Therefore, it is less affected by the zero-shot setting, which means that if the trainset includes objects from testset and the model has excellent local content learning capabilities, the model can still perform well on the testset.
>
> ii) Effectiveness of FineCLIP. According to Figure 2(a) in our paper, FineCLIP greatly outperforms RegionCLIP and CLIPSelf, both of which also utilize region information. This result indicates that the region-text pairs generated using YOLO and BLIP2 effectively cover objects in testset, and FineCLIP has superior local feature learning capabilities compared to competing methods.
>
> iii) Impact of Trainset Size. Additionally, we observe that when using the same scale of 100K training samples, FineCLIP under zero-shot setting does not surpass that of using in-domain COCO train2017 split in either task. However, as the data scale increases, FineCLIP achieves superior performance in box classification task, demonstrating FineCLIP's outstanding scalability.
>
>
> Q2: Will different region proposal methods have a big influence on the final results?
>
> A2: Thanks. Based on COCO Train2017 split, we evaluate four different region proposal methods: manual annotation $^{[1]}$, FastSAM $^{[2]}$, RPN $^{[3]}$ and YOLOv9 $^{[4]}$, to show their impact on the performance of FineCLIP. The results are detailed in Table 5 of the attached PDF document. From the obtained results, we find three key insights below.
>
> i) Automated vs. Manual Proposals.
> Automated region proposals yield results comparable to manually annotated high-quality regions. They slightly perform better in box classification and worse in retrieval tasks, proving the feasibility of automated methods.
>
> ii) Box Quantity: More boxes do not necessarily mean better performance. FastSAM generates the most boxes, but they appear too cluttered upon manual inspection, leading to poor model performance.
>
> iii) RPN offers a moderate number of region proposals, resulting in balanced performance. YOLOv9, focusing more on specific object categories, produces fewer but more precise boxes, achieving the best box classification performance.
>
> Overall, different region proposal methods indeed have a big influence on the final results. In this paper, we select YOLOv9 because of its efficiency and its significant enhancement of local feature extraction.
>
>
> References:
>
> [1] Microsoft COCO: Common Objects in Context.
>
> [2] Fast Segment Anything.
>
> [3] Faster R-CNN: Towards Real-Time Object Detection with Region Proposal Networks.
>
> [4] YOLOv9: Learning What You Want to Learn Using Programmable Gradient Information.

---

> > ### Comment · Reviewer_Ask3 · 2024-08-12
> > **Reseponse to rebuttal by authors**
> >
> > Thanks for the response. It addressed my questions. However, considering that the final results will be primarily influenced by the quality of region preparation, together with the concerns from Reviewer oDkS about resource consumption, Reviewer rzQ8 about the doubt of FineCLIP's ability to pre-training from scratch, I re-evaluated the limitation of the proposed method. I decided to lower the rating to 5, which shows that my evaluation of this work is reduced, but I have no problem with it being accepted.

---

> > > ### Author Response · Authors · 2024-08-12
> > > **Thanks for the response of Reviewer Ask3**
> > >
> > > Thank you for your valuable feedback. We are pleased to have addressed the concerns you previously raised and would like to further discuss the new points you've mentioned.
> > >
> > > Concern 1: The Influence of Region Proposals.
> > >
> > > As shown in Table 5 of the attached PDF document, the region proposal method indeed has a direct impact on FineCLIP's performance. However, as demonstrated in Table 8 in the Appendix of our paper, even with the least effective region proposal method, FineCLIP still significantly outperforms other competing methods.
> > >
> > > Concern 2: Resource Consumption.
> > >
> > > As shown in Table 1 of the attached PDF file, while FineCLIP does require high GPU memory usage, its training time is very similar to that of CLIPSelf and RegionCLIP.
> > > Therefore, we believe the training time required for FineCLIP is acceptable.
> > >
> > > Concern 3: FineCLIP's ability to pre-training from scratch.
> > >
> > > The primary focus of our paper is to validate the feasibility of FineCLIP in common fine-tuning settings. The extensive experiments presented in the paper and during the rebuttal phase are sufficient to demonstrate FineCLIP's superiority. While we did not conduct experiments to pre-train FineCLIP from scratch, the scalability experiments in Section 4.2 provide insight into FineCLIP's potential in this regard.
> > >
> > > We understand and respect your objective evaluation of our paper. However, we believe that the existing experiments and discussions adequately demonstrate the contributions of our work. Notably, after further discussion, Reviewer oDkS and Reviewer rzQ8 have both raised their scores. Given that your previous concerns have been addressed, we kindly request you to consider restoring your original score. We would greatly appreciate it.
> > >
> > > We sincerely appreciate your constructive comments and look forward to your reply. Thank you once again for your understanding and support of our work.

---

> > > > ### Comment · Reviewer_Ask3 · 2024-08-14
> > > > **A new concern about region proposal and region annotation**
> > > >
> > > > With the help of well-trained Yolov9 and LVLM, FineCLIP obtains bounding boxes for regions and the corresponding labels (text descriptions) and uses them in the fine-tuning of the pre-trained model. These annotations most likely include the region-category pairs in the subsequent OVD task, regardless of the novel or base categories. Therefore, the problem is that using detectors and an extra strong LVLM to gather training data is unfair since the detector might be trained on the novel classes in the final OVD test set, and the extra LVLM, like BLIP2, will give the class information to the region proposals generated by the detector. Then, the CLIP is being trained with those classes using more direct supervision, which is against the idea of OVD detection. We are not sure how much the performance increase is from that supervision and how much from the improved regional awareness ability of CLIP.
> > > >
> > > > RegionCLIP and CLIPSelf may not have the same problem. RegionCLIP uses RPN to get regions, while it relies on CLIP itself to match region and text descriptions. CLIPSelf also generates region proposals, while it distillates the region embedding from CLIP itself to its dense feature, and there are no new text annotations involved. Therefore, for these two methods, there are no direct supervision annotations that are related to the OVD test set and added to the original training data (CC3M) in the pre-trained model fine-tuning stage.

---

> > > > > ### Author Response · Authors · 2024-08-14
> > > > > **Response to Reviewer Ask3**
> > > > >
> > > > > Thank you for your thoughtful question. We'd like to address it from the following perspectives:
> > > > >
> > > > > i) As shown in Table 6 of the attached PDF, we also evaluated FineCLIP under conditions similar to those used for CLIPSelf$^{[1]}$.
> > > > > Specifically, we trained FineCLIP on COCO Train2017, using the region proposals provided by the dataset, with region annotations generated by BLIP2.
> > > > > As stated in the BLIP2 paper$^{[2]}$, BLIP2 was not trained on the OV-COCO and OV-LVIS datasets, so there is no issue of label leakage.
> > > > > In this fair comparison, FineCLIP outperforms CLIPSelf on both $AP_{50}^{base}$ and $AP_{50}$, even with lower resolution inputs.
> > > > >
> > > > > ii) FineCLIP's strength lies in its ability to learn from training data more effectively.
> > > > > As you noted, CLIPSelf relies on a pre-trained frozen teacher model, while RegionCLIP depends on pseudo-labels generated by CLIP.
> > > > > We understand your concerns, but the inability of these two methods to fully leverage region-text data is their limitation, not our fault.
> > > > > In contrast, FineCLIP not only distills region embeddings to local dense features but also acquires knowledge from training data through the multi-grained contrastive learning paradigm.
> > > > >
> > > > > We hope the above responses could address your concerns. Please reach out if there is any confusion. Thank you very much.
> > > > >
> > > > > Reference:
> > > > >
> > > > > [1] CLIPSelf: Vision Transformer Distills Itself for Open-Vocabulary Dense Prediction
> > > > >
> > > > > [2] BLIP-2: Bootstrapping Language-Image Pre-training with Frozen Image Encoders and Large Language Models.

---

### Official Review · Reviewer_oDkS · 2024-07-11

**Soundness:** 3
**Presentation:** 4
**Contribution:** 2
**Rating:** 5
**Confidence:** 4

**Summary:**

The paper introduces FineCLIP, a novel vision-language model designed to enhance fine-grained understanding in image-text tasks. It addresses limitations in existing models like CLIP, which struggle with dense prediction tasks due to a lack of fine-grained detail comprehension. The authors propose two main innovations: A real-time self-distillation scheme that transfers representation capabilities from global to local image features, facilitating a deeper understanding of local image details. A semantically-rich regional contrastive learning approach using generated region-text pairs to boost local representation capabilities with fine-grained knowledge.

**Strengths:**

The paper introduces FineCLIP, a novel vision-language model designed to enhance fine-grained understanding in image-text tasks. It addresses limitations in existing models like CLIP, which struggle with dense prediction tasks due to a lack of fine-grained detail comprehension. The authors propose two main innovations: A real-time self-distillation scheme that transfers representation capabilities from global to local image features, facilitating a deeper understanding of local image details. A semantically-rich regional contrastive learning approach using generated region-text pairs to boost local representation capabilities with fine-grained knowledge.

**Weaknesses:**

I understand that the author discusses the differences and experimental comparisons with existing methods such as RegionCLIP and CLIPSelf. However, in my opinion, the self-distillation scheme is a very common practice, and its contribution is not particularly novel. The semantically-rich regional contrastive learning paradigm is a significantly effective method for aligning image local regions and text. However, sending all segmented local images into the vision-language model (VLLM) to generate text descriptions does not seem very efficient, especially when scaled to larger datasets.

The paper could provide more details on the computational efficiency of FineCLIP, especially the training cost, and a comparison with an existing model (CLIPSelf).

Regarding the ablation study in Table 1, the author should further explain why using only $L_{SD}$ for supervision leads to model collapse. Additionally, the effect of using $L_RC$ alone should be supplemented.

In the open-vocabulary semantic segmentation task shown in Table 4, CatSeg+FineCLIP shows only a minor improvement compared to CatSeg+CLIPSelf. This does not sufficiently support the author's claim of fine-grained semantic alignment.

Importantly, if my understanding is incorrect, please correct me, and I will improve my score accordingly.

**Questions:**

Minor  question

The authors highlight the 'real-time' capability of the self-distillation scheme. Could you please clarify the significance of this feature? Is real-time operation essential for the effectiveness of your method or does it offer specific advantages in the given context?

**Limitations:**

Yes, the authors addressed limitations, potential negative societal impact, and mitigation.

---

> ### Author Rebuttal · Authors · 2024-08-07
>
> Q1: As a common practice, the contribution of self-distillation scheme is not novel. What are significance and advantages of "real-time" capability of the self-distillation scheme? Why does using only $L_{SD}$ for supervision lead to model collapse?
>
> A1: We fully understand your concerns. To better illustrate the novelty of FineCLIP, we will answer the above questions together.
>
> Self-distillation is indeed a common practice, typically involving the distillation of some capabilities from a frozen teacher model to a trainable student model, as done in CLIPSelf. However, this implementation is only suitable for further fine-tuning of CLIP, and fundamentally conflicts with multi-modal pre-training scenarios. This conflict mainly arises from two aspects: i) This implementation requires a pre-trained teacher model as a prerequisite. ii) The frozen teacher model limits the performance ceiling of the student model, whereas the goal of pre-training is to continuously improve the model. Given that self-distillation has shown great potential in enhancing local feature extraction abilities, the value of our real-time self-distillation scheme lies in effectively integrating the self-distillation strategy into the pre-training process, fully leveraging its advantages to build a more powerful pre-trained model.
>
> In terms of implementation, our real-time self-distillation scheme enables a trainable model to act both as the teacher and the student, achieving true "self-distillation". The novelty of this scheme is that it resolves the conflicts mentioned earlier: i) This scheme does not require a frozen teacher model, allowing the model to realize self-guidance. ii) During the pre-training process, the model continuously improves its global representation capabilities, thus more effectively guiding local feature extraction and breaking through performance ceilings, embodying the value of "real-time". We have demonstrated the effectiveness, stability, and scalability of FineCLIP through extensive experiments.
>
> It is important to note that this scheme is tailored for multi-modal pre-training and relies on global contrastive learning (also denoted as "real-time" in our paper) to work effectively. When only $L_{SD}$ is used for supervision, the lack of global contrastive learning to maintain the model's global representation capabilities leads to the degradation of semantic meaning in model's global embeddings and local features. Consequently, the uni-modal alignment enforced by $L_{SD}$ becomes meaningless, ultimately causing the model collapse.
>
>
> Q2: Sending all segmented local images into VLLM to generate text descriptions does not seem very efficient, especially when scaled to larger datasets.
>
> A2: Good question. We answer it from the following three aspects:
>
> i) Demand for data scale and quality. In the era of large models, the scale and quality of data have been proven to be critical factors affecting the performance of pre-trained models$^{[1]}$. Therefore, expanding the data scale and improving the data quality is an inevitable trend. Compared to manual annotation, using LLM or VLLM for automated data generation is much more efficient.
>
> ii) Successful cases of large-scale data annotation with large models. In the field of LLM, using LLMs for large-scale pre-training data synthesis and cleaning has become a standard practice$^{[2,3]}$. In the contrastive learning domain, LaCLIP$^{[4]}$ utilizes LLM to augment the textual content of LAION-400M dataset$^{[5]}$, processing data on a scale larger than that involved in our work.
>
> iii) Efficiency of data annotation with large models. In our response to common questions, we show the time cost of data preparation. Given the relatively small amount of data used in our experiment, further optimizations on the inference speed are not performed. However, deploying models using the vLLM$^{[6]}$ framework could significantly increase the throughput of data generation, potentially improving processing speed by at least 20 times.
>
> In summary, using LVLM for large-scale data annotation is not only feasible but also advantageous in efficiency.
>
> Q3: As shown in Table 4, CatSeg+FineCLIP shows only a minor improvement compared to CatSeg+CLIPSelf, which is not sufficiently support the author's claim of fine-grained semantic alignment.
>
> A3: We appreciate the opportunity to clarify the performance of FineCLIP.
> We believe that the improvements brought by FineCLIP to CatSeg are comprehensive and significant, adequately demonstrating FineCLIP's superiority in fine-grained semantic understanding. Please see our detailed explanations blow.
>
> On one hand, according to Table 4 in our paper, CatSeg+FineCLIP surpasses CatSeg+CLIPSelf on all metrics. On the other hand, we calculate the average improvements brought by CLIPSelf and FineCLIP to CatSeg on mIoU and mAcc across three benchmarks for straightforward comparisons, and show the results in Table 7 of the attached PDF document. We can find that the improvements provided by FineCLIP significantly exceed those brought by CLIPSelf, particularly when using the ViT-B/16 backbone.
>
> Q4: The effect of using $L_{RC}$ alone should be supplemented.
>
> A4: Thanks for your suggestion. We present the results of using $L_{RC}$ alone in line \#4 of Table 4 in the attached PDF document.
> We can observe that regional contrastive learning benefits the model's local feature extraction abilities and can effectively combine with global contrastive learning and self-distillation scheme to obtain the optimal fine-grained understanding performance.
>
> References:
>
> [1] Scaling Laws for Neural Language Models.
>
> [2] The Llama 3 Herd of Models.
>
> [3] Qwen2 Technical Report.
>
> [4] Improving CLIP Training with Language Rewrites.
>
> [5] Open Dataset of CLIP-Filtered 400 Million Image-Text Pairs.
>
> [6] Efficient Memory Management for Large Language Model Serving with Paged Attention.

---

> > ### Comment · Reviewer_oDkS · 2024-08-12
> > **Thanks for the rebuttal**
> >
> > The rebuttal has addressed most of my concerns. However, I still have concerns regarding the resource consumption of the proposed approach relative to its performance improvement. Specifically, I notice that CatSeg+FineCLIP does not demonstrate a significant advantage over CatSeg+CLIPSelf when using ViT-L/14 as the backbone. Could the authors provide a clear overview of the resource consumption? Additionally, could the authors briefly explain why FineCLIP shows more substantial improvement in open-vocabulary object detection? Will the data and code be made publicly available?

---

> > > ### Author Response · Authors · 2024-08-12
> > > **Replying to Reviewer oDkS**
> > >
> > > Thank you for your valuable feedback. Below are our responses to your additional questions.
> > >
> > > Q1: Compared to using ViT-B/16 as the backbone, CatSeg+FineCLIP does not demonstrate the same significant advantage over CatSeg+CLIPSelf when using ViT-L/14 as the backbone.
> > >
> > > A1: This is an insightful observation. We believe that one important reason for this is that CatSeg with ViT-L/14 already exhibits much better performance than with ViT-B/16, thereby making further improvements more difficult to achieve. Nonetheless, FineCLIP still provides more remarkable enhancements to CatSeg than CLIPSelf when using ViT-L/14.
> > >
> > >
> > > Q2: Could the authors provide a clear overview of the resource consumption?
> > >
> > > A2: We detail the training time cost and GPU memory usage in Table 1 of the attached PDF document. We fully understand your concerns. Due to FineCLIP incorporating the multi-grained contrastive learning paradigm and the self-distillation scheme, it requires relatively higher GPU memory usage. However, it is important to note that the per-epoch training time cost for FineCLIP (11 minutes) is very close to that of CLIPSelf (10 minutes) and RegionCLIP (9 minutes). Thus, the training time required for FineCLIP is entirely acceptable.
> > >
> > > Furthermore, we also present the details of the data preparation in A2 of the common responses. The construction of CC2.5M trainset can be completed within a single day.
> > >
> > >
> > > Q3: Could the authors briefly explain why FineCLIP shows more substantial improvement in open-vocabulary object detection tasks?
> > >
> > > A3: Good question. We believe there are three main reasons.
> > >
> > > i) The semantic information contained in visual dense features is crucial for performance in open-vocabulary object detection tasks. FineCLIP successfully enhances the accuracy and richness of the semantic representation of visual dense features through regional contrastive learning and self-distillation scheme.
> > >
> > > ii) The CC2.5M trainset we constructed provides abundant and valuable fine-grained semantic knowledge.
> > >
> > > iii) Compared to CLIPSelf, FineCLIP learns the knowledge from training data more effectively. While that CLIPSelf learns only from the region embeddings derived from a frozen teacher model, FineCLIP not only learns from region embeddings that can be improved during the training but also achieves fine-grained region-text alignment through regional contrastive learning.
> > >
> > > Q4: Will the data and code be made publicly available?
> > >
> > > A4: Yes, we will open-source the data, code and model weights immediately if our paper is accepted.
> > >
> > > We hope the above clarifications could address your concerns. We are looking forward to your feedback if there is still any confusion. Thank you very much again for your insightful comments.

---

> > > > ### Comment · Reviewer_oDkS · 2024-08-12
> > > > **Thanks again for the additional response**
> > > >
> > > > I notice that the GPU memory usage for FineCLIP is indeed higher compared to the previous method, while the training time is acceptable. I'll raise my rating to a BA.

---

> > > > > ### Author Response · Authors · 2024-08-12
> > > > > **Thanks for the response of Reviewer oDkS**
> > > > >
> > > > > We sincerely thank you for raising your score. We greatly appreciate your constructive comments. We will carefully incorporate the key points of our discussion into the revision. Thank you very much again!

---

### Official Review · Reviewer_5tL8 · 2024-07-12

**Soundness:** 2
**Presentation:** 3
**Contribution:** 3
**Rating:** 5
**Confidence:** 3

**Summary:**

To address CLIP's limitations in understanding fine-grained details, the authors propose FineCLIP, a method for training CLIP-based architectures that proposes two novel losses, a real-time self-distillation loss and a regional contrastive loss. The regional contrastive loss is designed to encourage learning of fine-grained semantic features by querying a Large Vision-Language Model (LVLM) with region crops for a detailed description, then enforcing a contrastive loss between the corresponding text embedding and region's pooled visual embedding. The real-time self-distillation loss encourages pooled regional vision encoder features to be consistent with the vision encoder's embedding of region crops and is designed to prevent loss of global features during training with the regional contrastive loss. To assess the performance of FineCLIP, the authors conduct experiments on Open-Vocabulary Object Detection, Open-Vocabulary Semantic Segmentation and Zero-Shot Image-Text Retrieval and find FineCLIP performs favorably compared to the baselines trained and evaluated in the same setting. They perform ablation studies to assess the contributions of FineCLIP's losses, the region annotation method, and impact of the train dataset size.

**Strengths:**

1. The writing is easy to follow.
2. The FineCLIP method consists of losses that are straightforward to implement and can be trained end-to-end.
3. The ablation studies clearly show the contribution of each component of FineCLIP to its performance and justify the choice of region annotation method. The accompanying analyses, including the study of dataset size, are detailed and insightful.

**Weaknesses:**

1. Trainability: It is difficult to tell how usable FineCLIP is without details on the time it takes to train a single epoch and the GPU memory the method uses. How big is the LVLM? Is it queried on the fly or ahead of time? How long does inference take? The authors should include this information for FineCLIP and CLIPSelf to demonstrate the method's ease of use during training and inference.
2. Analysis of results: The authors provide a short explanation that the mixed results of FineCLIP in Table 4 on PC-59 and ADE-847 are due to the data distribution gap between CC2.5M and Pascal Context, but it would be better to provide more of a semantic analysis in which classes are underperforming and why this is the case. This would help strengthen the analysis of FineCLIP's limitations.

**Questions:**

1. Under what conditions does adding $\mathcal{L}_{SD}$ help? It's confusing why all three losses in row 5 of Table 1 don't give the best retrieval performance.
2. In l. 26 what are the domains specifically in the "domain shift"?
3. In l. 48 what losses are happening during the pretraining stage?
4. In l. 108-109 it says "instance-level" alignment but does this instance refer to a single image not an instance of an object?
7. In Table 2, what is the time required for each region annotation method?
8. In section 4.2, the smallest dataset is 100k samples. However, often in the case of dense prediction tasks, there is a very limited amount of training data. Why did the authors choose to scale the dataset down rather than up and see how it affects performance?
9. Can the authors explain the performance discrepancy between CLIPSelf in Table 3 in the original paper [1] versus this work?

Minor Typos/Suggestions:
- l. 22 remove "been"
- l. 134 "learn" <- is this right?
- l. 344 "realize"
- l. 285 "detention" <- "detection"
- l. 228 "We" is capitalized
- l. 262 "quantitative"
- l. 272 "trainset set"
- l. 285 "detention"

[1] Size Wu, Wenwei Zhang, Lumin Xu, Sheng Jin, Xiangtai Li, Wentao Liu, and Chen Change Loy. CLIPSelf: Vision transformer distills itself for open-vocabulary dense prediction. In CVPR, 2024.

**Limitations:**

The authors have included a limitations section on the scalability of FineCLIP to larger datasets, but might improve this section with an analysis of how FineCLIP performs with smaller datasets, and a more detailed semantic analysis of when FineCLIP does not outperform previous methods.

---

> ### Author Rebuttal · Authors · 2024-08-07
>
> Q1: The authors could provide more semantic analysis to better explain the performance of FineCLIP on downstream tasks.
>
> A1: Following your suggestion, we add semantic distribution statistics in Table 1 of the attached PDF document.
> The results show that the training data in CC2.5M related to the labels in PC-59 has the lowest proportion, which explains why training on CC2.5M does not significantly improve model performance on PC-59.
> Furthermore, ADE-847 has more than half of all 847 categories being irrelevant or only weakly relevant to CC2.5M, making it the most challenging benchmark.
>
> Q2: Under what conditions does add $L_{SD}$ help? It's confusing why all three losses in row 5 of Table 1 don't give the best retrieval performance.
>
> A2: According to the results shown in Table 1 (line \#2 vs \#3, line \#4 vs \#5) of our paper, $L_{SD}$ improves local feature extraction but slightly reduces retrieval performance.
>
> Importantly, we conduct further validation by using trainset in different scales, with results shown in Table 3 of the attached PDF document.
> We observe an interesting trend: as the scale of trainset increases, the negative impact of $L_{SD}$ on retrieval tasks gradually diminishes.
> This means that $L_{SD}$ tends to have a positive impact on retrieval performance when FineCLIP is fine-tuned with larger-scale data.
>
> Q3: Why did the authors choose to scale the dataset down rather than up and see how it affects performance?
>
> A3: Due to the limitations in computational resources, we could only afford to experiment with a total of 2.5M training samples. With our current experiment setup, using a single server with 8xA800 GPUs, the experiments in Section 4.2 alone would take at least two weeks. If we increase the training samples from \{100K, 500K, 2.5M\} to \{1M, 5M, 25M\}, the experimental duration will extend to at least two months, which is far beyond our capacity.
>
> Q4: Can the authors explain the performance discrepancy between CLIPSelf in Table 3 in the original paper versus this work?
>
> A4: The training image resolution is the key factor of this performance discrepancy. Specifically, in our paper, the training image resolution was standardized to 224 for ViT/B and 336 for ViT/L, whereas the original CLIPSelf used a resolution of 1024. CLIPSelf, an efficient distillation method based on pre-trained CLIP, allows the input image resolution to be increased to 1024 due to its low computational costs. According to Table 1(c) in CLIPSelf paper $^{[1]}$, increasing resolution from 320 to 1024 boosts Top 1 Mean Accuracy by 25.6\%. However, contrastive per-training methods have much higher GPU memory costs, requiring a trade-off between batch size and input image size, making it challenging to achieve such high resolutions. To ensure a fair comparison, we used the default image size employed by CLIP, resulting in a performance decline for CLIPSelf in this work compared to the original paper.
>
> Q5: What are the domains specifically in the "domain shift" in Line 26?
>
> A5: The term "domain shift" in Line 26 follows the usage in RegionCLIP paper $^{[2]}$. It means that CLIP aligns global image contents with corresponding global text descriptions, but cannot realize well fine-grained alignment between local image regions and corresponding local textual concepts. This lack of fine-grained alignment results in suboptimal performance on downstream tasks requiring fine-grained abilities. We understand your concern that "domain shift" might be too abstract in this context. "Task shift" might be a more accurate expression.
>
> Q6: What losses are happening during the pre-training stage in Line 48?
>
> A6: The "pre-training" mentioned in Line 48 refers to the training process of FineCLIP, not the pre-training of CLIP. Therefore, the loss function at this stage is the complete loss function of FineCLIP as defined in Equation 5. We describe it in this way because FineCLIP is essentially a pre-training method based on contrastive learning combined with self-distillation scheme.
>
> Q7: Does the "instance" in Line 108-109 refer to a single image or an instance of object?
>
> A7: The "instance" in Line 108-109 refers to the image and corresponding text.
>
> References:
>
> [1] CLIPSelf: Vision Transformer Distills Itself for Open-Vocabulary Dense Prediction.
>
> [2] RegionCLIP: Region-based Language-Image Pretraining.

---

> > ### Comment · Reviewer_5tL8 · 2024-08-09
> > **Typo in my question 6**
> >
> > Sorry, my question was not clear. 100k samples is very large for a dense prediction dataset. E.G. Cityscapes has <3k finely labeled samples. Do you have any insight into how FineCLIP performs when there are only several thousand samples in the dataset?

---

> > > ### Author Response · Authors · 2024-08-10
> > > **Additional experiments with small-scale trainset**
> > >
> > > Thank you for your suggestion. Following your advice, we carefully conduct additional experiments to investigate the performance of FineCLIP with a small-scale trainset. Specifically, we randomly select 1,000 data samples from CC2.5M as our trainset and evaluate the performance of methods on the COCO val2017 split. The results are summarized in the Rebuttal Table 1 below.
> > >
> > > Rebuttal Table 1: Zero-shot performance comparisons with 1K training samples.
> > >
> > > | #   | Method           | Box |        Classification           | Retrieval |           |
> > > |-----|------------------|--------------------|------------------|-----------|-----------|
> > > |     |                  | Top1               | Top5             | I2T       | T2I       |
> > > | 1   | CLIP             | 30.2               | 52.3             | 51.5      | 35.8      |
> > > | 2   | RegionCLIP       | 35.0               | 62.1             | 19.4      | 13.1      |
> > > | 3   | CLIPSelf         | 37.9               | 66.2             | 35.2      | 28.4      |
> > > | 4   | FineCLIP (Ours)  | 42.0               | 66.8             | 49.2      | 36.0      |
> > >
> > > As shown in the table, FineCLIP effectively enhances CLIP's fine-grained understanding capability, even with a very small amount of training data (1K samples). Moreover, FineCLIP still outperforms other competing methods by providing the most significant improvement in box classification task while maintaining strong retrieval ability.
> > >
> > > We hope these experiments could address your concerns. We greatly appreciate your insightful feedback and are happy to provide further clarification if needed. Thank you very much again.

---

> > > > ### Comment · Reviewer_5tL8 · 2024-08-13
> > > > **Response to Rebuttal**
> > > >
> > > > Thanks to the authors for their comprehensive response. While my concerns about the semantic analysis of FineCLIP's performance and the performance at smaller dataset scales has been addressed, the computational efficiency of the method is still concerning: the memory requirement of the method may require a multi-gpu implementation and my hope is that the authors release code that supports this.

---

> > > > > ### Author Response · Authors · 2024-08-13
> > > > > **Response to Reviewer 5tL8**
> > > > >
> > > > > Thank you for your response. We are pleased to see that your concerns have been addressed.
> > > > >
> > > > > Regarding the multi-gpu training code, we have included it in the supplementary material and will open-source the code as soon as our paper is accepted.
> > > > >
> > > > > We notice that you lowered the Rating. If there are any remaining questions or concerns, we'd be happy to address them. We look forward to your reply. Thank you.

---

> > > > > > ### Comment · Reviewer_5tL8 · 2024-08-13
> > > > > > **Response to rebuttal**
> > > > > >
> > > > > > My concern on semantic analysis has been addressed but my concern on the usability of the method has not been assuaged because of FineCLIP's large memory requirements in row 5 Table 1 of the rebuttal pdf. Such a large memory requirement may be prohibitory. One question I have is whether the authors believe it can be reduced.

---

> > > > > > > ### Author Response · Authors · 2024-08-14
> > > > > > > **Response to Reviewer 5tL8**
> > > > > > >
> > > > > > > Thanks for your feedback. We understand your concerns regarding the GPU memory usage.
> > > > > > >
> > > > > > > First,  we believe that using frameworks such as DeepSpeed or FlashAttention can significantly reduce the GPU memory consumption, which is a widely adopted strategy in the training of large models.
> > > > > > >
> > > > > > > Also, as you have mentioned, contrastive learning is usually conducted in multi-GPU environments, which can easily meet FineCLIP's GPU memory requirements.
> > > > > > >
> > > > > > > We hope this response could address your concerns. Thank you very much again.

---

### Author Rebuttal · Authors · 2024-08-07

We'd like to thank all the reviewers for the valuable comments and suggestions. We will respond to common questions in this general rebuttal.

Q1: The authors should provide more information of the training time cost and GPU memory usage to demonstrate the ease of FineCLIP.

A1: Good suggestion. We present these details in Table 1 of the attached PDF document. As a multi-grained contrastive learning method, FineCLIP requires relatively high GPU memory usage, but its training time cost is very close to that of CLIPSelf and RegionCLIP under the same setting.

Q2: Questions about CC2.5M construction. i) What is the criteria for selecting CC2.5M from CC3M? ii) Are the region information prepared before the CLIP fine-tuning or real-time processing during the training? iii) What about the time cost of CC2.5M construction?

A2: i) The image selection process consists of two steps. To begin with, we use the YOLOv9 to propose regions for all images in CC3M, discarding images that do not yield any region proposals. Then, we filter out the images with low resolution, which results in 2.5 million selected images.
ii) The region proposals and annotations are prepared before training, rather than being processed in real-time. This is due to two drawbacks of real-time processing: a reduction in training speed, and the randomness of region annotations (which compromises the comparison fairness).
iii) We utilize 8xA800 GPUs for data preparation. It takes 4.5 hours to generate 10.4 million region proposals using YOLOv9 and 12.5 hours to caption these regions with BLIP2-COCO-6.7B.

---

### Decision · Program_Chairs · 2024-09-25

**Decision:**

Accept (poster)

**Comment:**

This paper presents a method called Fine VLIP for understanding fine-grained details. The authors proposed two novel losses to consider real-time self-distillation and regional contrastive learning. The validity of the proposed method is demonstrated through detailed experiments by showing superior performance to the previous works in most cases. Ablation study also demonstrates the importance of all the key components. The reviewers’ scores are consistently positive.

Here are just a few comments to improve this paper:
- The authors are advised to make the texts in the figures and tables much larger, e.g., as large as those in the main texts.
- “log” in (4) should be “\log”. Similarly, “exp” should be “\exp”.
- “We” in line 228 should be “we”.